# Double Descent and Other Interpolation Phenomena in GANs

## Abstract

We study overparameterization in generative adversarial networks (GANs) that can interpolate the training data. We show that *overparameterization can improve generalization performance and accelerate the training process*. We study the generalization error as a function of latent space dimension and identify two main behaviors, depending on the learning setting. First, we show that overparameterized generative models that learn distributions by minimizing a metric or $f$-divergence do not exhibit double descent in generalization errors; specifically, *all* the interpolating solutions achieve the same generalization error. Second, we develop a novel pseudo-supervised learning approach for GANs where the training utilizes pairs of fabricated (noise) inputs in conjunction with real output samples. Our pseudo-supervised setting exhibits double descent (and in some cases, triple descent) of generalization errors. We combine pseudo-supervision with overparameterization (i.e., overly large latent space dimension) to accelerate training while matching or even surpassing generalization performance without pseudo-supervision. While our analysis focuses mostly on linear models, we also apply important insights for improving generalization of nonlinear, multilayer GANs.

## 1 Introduction

Generative adversarial networks (GANs) (Goodfellow et al., 2014) are a prominent concept for addressing data generation tasks in contemporary machine learning. GANs learn a data generator model that produces new instances from a data class represented by a set of training examples. A GAN's generator network is trained in conjunction with a discriminator network that evaluates the generator's ability and directs it towards better performance. GANs have an intricate design and training philosophy whose theory and practice are still far from being sufficiently understood.

A key aspect that complicates the understanding of GANs is that, like many other deep learning architectures, they are highly complex models with typically many more parameters than the number of training data samples. Therefore, GANs are *overparameterized models* that can be trained to interpolate (i.e., memorize) their training examples. Yet, overparameterized GANs are capable of generating high quality data beyond their training datasets. The analysis of overparameterized machine learning is a highly active research area that is mainly focused on supervised learning problems such as regression (Belkin et al., 2019b; Bartlett et al., 2020; Muthukumar et al., 2020b; dAscoli et al., 2020) and classification (Muthukumar et al., 2020a; Deng et al., 2019; Kini & Thrampoulidis, 2020). *The study of overparameterization in the unsupervised learning and data generation problems relevant to GANs is uncharted territory that we are first to explore in this paper.*

This paper develops a new framework for the study of generalization and overparameterization in GANs and PCA. *We examine the generalization of linear GANs at different parameterization levels by varying the latent space dimension*, which in a GAN is the dimension of the input (random noise) vectors to the data generator. This is a practical way of controlling the parameterization of our models, since we do not need to consider modifying the width or depth of the generator network. Our framework leads us to the following

key insights on how the generalization performance of overparameterized linear GANs is affected by the training approach.

First, GAN training via **minimization of a distribution metric or $f$-divergence** results in *unsatisfactory generalization performance* when the generator model is overparameterized and interpolates its noisy training data. Specifically, we prove that under such a training process *all* overparameterized solutions have the same generalization performance. Moreover, the best generalization is obtained by an underparameterized solution with the same dimension as the true latent space dimension of the data, which is usually unknown. This set of interpolating solutions which have constant test error establishes a new generalization behavior of generative models.

Second, our theoretical studies inspire a new **pseudo-supervised training** regime for GANs and show that it can *improve generalization performance in overparameterized settings* where interpolation of noisy training data occurs. Our pseudo-supervised approach selects a subset (or all) of the training data examples and individually associates them with random (noise) vectors that act as their latent representations (i.e., the inputs given to the generator to yield the respective training data). Pseudo-supervision *accelerates the training process* and improves generalization by reducing the number of effective degrees of freedom in overparameterized GAN learning (although in many cases the learned GAN can still interpolate the training data). We develop several implementations for the pseudo-supervised optimization objective and examine their respective generalization behaviors, which we show to include *double descent* and also *triple descent* of generalization errors as a function of the latent space dimension of the learned GAN.

Third, encouraged by our new insights into linear GANs, we explore their implications for *nonlinear, multilayer* GANs. Specifically, we implement and study our pseudo-supervised learning scheme for a gradient-penalized Wasserstein GAN (Gulrajani et al., 2017) on the MNIST digit dataset of binary images. Our results demonstrate that pseudo-supervised learning significantly improves generalization performance and accelerates training when compared to training the same GAN without pseudo-supervision.

## 2 Related work

GANs (Goodfellow et al., 2014) have been very successful in modeling complex data distributions, such as distributions of images (Brock et al., 2018; Karras et al., 2019a;b). These models are usually trained by having two competing networks: a generator network which attempts to approximate the data distribution and a discriminator network which attempts to classify between data from the training set and generated data. The objective function can either be an $f$-divergence (Goodfellow et al., 2014; Nowozin et al., 2016; Sarraf & Nie, 2021) or a metric (Arjovsky et al., 2017; Gulrajani et al., 2017) and is typically minimized by the generator while simultaneously being maximized by the discriminator. This minmax game can be unstable (Salimans et al., 2016; Mescheder et al., 2018) and is hard to analyze in full generality; therefore we turn to linear GANs.

Feizi et al. (2020) have studied GANs with linear generators, quadratic discriminators, and Gaussian data (this has been named the LQG setting).

In this setting, the objective loss is the 2-Wasserstein distance between two Gaussian distributions $\mathcal{N}(\boldsymbol{\mu}_1, \boldsymbol{\Sigma}_1)$ and $\mathcal{N}(\boldsymbol{\mu}_2, \boldsymbol{\Sigma}_2)$:

$$\mathcal{W}_2^2(\mathcal{N}(\boldsymbol{\mu}_1, \boldsymbol{\Sigma}_1), \mathcal{N}(\boldsymbol{\mu}_2, \boldsymbol{\Sigma}_2)) = \|\boldsymbol{\mu}_1 - \boldsymbol{\mu}_2\|_2^2 \tag{1}$$
$$+ \mathrm{Tr}(\boldsymbol{\Sigma}_1) + \mathrm{Tr}(\boldsymbol{\Sigma}_2) - 2\mathrm{Tr}\left(\left(\boldsymbol{\Sigma}_1^{\frac{1}{2}} \boldsymbol{\Sigma}_2 \boldsymbol{\Sigma}_1^{\frac{1}{2}}\right)^{\frac{1}{2}}\right).$$

This distance is well known (Givens et al., 1984; Olkin & Pukelsheim, 1982) and is even used in the calculation of the well known evaluation metric FID (Heusel et al., 2017) in the GAN literature. One result in the LQG setting (Feizi et al., 2020) is that the principal component analysis (PCA) solution is an optimal solution for the generator in the minmax optimization.

In supervised problems, it was widely believed that the generalization error behavior as a function of the learned model complexity is completely characterized by the bias-variance tradeoff, i.e., in a supervised setting, the test error goes down and then back up as the learned model is more complex (e.g., has more

parameters). Relatively recently, it has been shown that test errors can have a *double descent* shape (Spigler et al., 2018; Belkin et al., 2019a) as a function of the learned model complexity. Specifically, in the double descent shape the test error goes back down when the learned model is sufficiently complex (i.e., overparameterized) to interpolate the training data (i.e., achieve zero training error). Remarkably, the double descent shape implies that the best generalization performance can be achieved despite perfect fitting of noisy training data. Typically, when models have many more parameters than training data, many mappings can be learned to perfectly fit (i.e., interpolate) the supervised pairs of examples. Therefore, a mapping with small norm is a natural (parsimonious) choice and tends to yield low test error even when the number of parameters is large. The research on overparameterized learning and double descent phenomena has been mostly focused on regression (Belkin et al., 2019b; Bartlett et al., 2020; Muthukumar et al., 2020b; dAscoli et al., 2020) and classification (Muthukumar et al., 2020a; Deng et al., 2019; Kini & Thrampoulidis, 2020) problems. Some work has been done in overparameterized GANs (Balaji et al., 2021) to understand how training stability is affected by increasing the width and depth of networks. By contrast, *we are the first to study generalization performance and double descent behavior in GANs.*

Since linear GANs are associated with PCA, this study relates to work on overparameterization in PCA (Dar et al., 2020) showing that, as one relaxes the orthonormal constraints and adds supervision to PCA, double descent emerges. Moreover, if the learning is fully supervised and has no orthonormal constraints, then the problem becomes linear regression that estimates a linear subspace. Hence, one can solve learning problems that are partly supervised and partly orthonormally constrained to obtain solutions to problems that are in-between PCA and linear regression. We will leverage this powerful idea to study overparameterization in linear GANs.

## 3 Bad generalization: Test errors are constant in the overparameterized regime

### 3.1 No double descent in generative models that minimize a metric or $f$-divergence

The goal of training GANs and generative models in general is to learn the distribution of the data. This is typically done by minimizing a distance between a fixed (i.e., given) distribution $p_f$, such as the empirical distribution of the training data, and the generated distribution $p_{\boldsymbol{\theta}}$ with parameters $\boldsymbol{\theta}$. The training dataset $\mathcal{D}$ includes $n$ examples $\{\mathbf{x}_i\}_{i=1}^n \in \mathbb{R}^d$. The next observation characterizes interpolating solutions for these kinds of problems.

**Observation 1.** *Let $P$ be the set of all probability distributions defined on the measurable space $(\Omega, \mathcal{F})$ equipped with any metric or f-Divergence denoted $q$. We let our training loss be $\mathcal{L}^{train}(\{\mathbf{x}_i\}_{i=1}^n, \boldsymbol{\theta}) = q(p_f, p_{\boldsymbol{\theta}})$ for $p_f, p_{\boldsymbol{\theta}} \in P$ and the test error is given by $\mathcal{L}^{test}(\boldsymbol{\theta}) = q(p_t, p_{\boldsymbol{\theta}})$ for the true distribution $p_t \in P$. Then, for any interpolating solution $\boldsymbol{\theta}$, i.e., any $\boldsymbol{\theta}$ so that $\mathcal{L}^{train}(\{\mathbf{x}_i\}_{i=1}^n, \boldsymbol{\theta}) = 0$, we have that*

$$\mathcal{L}^{test}(\boldsymbol{\theta}) = L_{interpolate}^{test}$$

*where $L_{interpolate}^{test}$ is a non-negative constant that depends on $q$ and $p_f$. Moreover, $p_{\boldsymbol{\theta}}$ is unique.*

*Proof.* Let $\boldsymbol{\theta}^*$ and $\boldsymbol{\theta}$ be two interpolating solutions. Since $q$ is a metric or $f$-divergence, the zero training errors of the interpolating solutions $\boldsymbol{\theta}^*$ and $\boldsymbol{\theta}$ imply that $p_{\boldsymbol{\theta}^*} = p_f$ and $p_{\boldsymbol{\theta}} = p_f$, implying uniqueness since $p_{\boldsymbol{\theta}} = p_{\boldsymbol{\theta}^*}$. Additionally,

$$\mathcal{L}^{\text{test}}(\boldsymbol{\theta}) = q(p_t, p_{\boldsymbol{\theta}}) = q(p_t, p_f) = q(p_t, p_{\boldsymbol{\theta}^*}) = \mathcal{L}^{\text{test}}(\boldsymbol{\theta}^*).$$

By letting $L_{\text{interpolate}}^{\text{test}} \triangleq q(p_t, p_f) \geq 0$, we get the desired result. $\qquad\square$

**Corollary 1.** *There is no double descent in generative models that minimize a metric or $f$-divergence, e.g., PCA for subspace learning, Jensen-Shannon GANs, WGANs, etc.*

In other words, there is no double descent behavior because the test error is *constant* in the overparameterized regime of interpolating solutions. This differs from the widely studied regression setup in that here we are trying to minimize the distance between two distributions rather than data points drawn from those

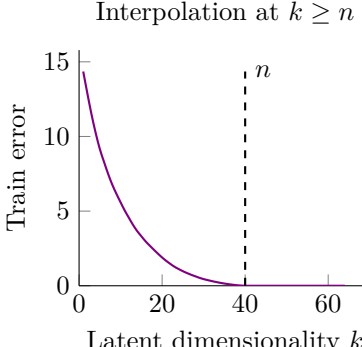
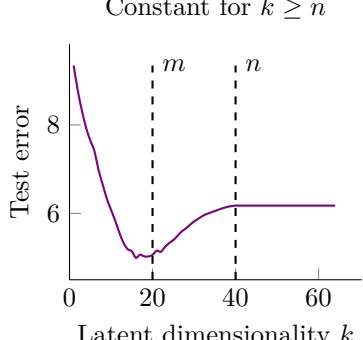

**Figure 1:** PCA and linear GAN's test error becomes constant when the model interpolates, i.e., when the latent dimensionality $k$ equals the number of training samples $n$. Therefore, the overparameterized regime does not exhibit double descent but rather a constant error. The test error achieves its minimum when the latent dimensionality $k$ is near the true model's dimensionality $m$. The train errors (left subfigure) and test errors (right subfigure) are calculated with the 2-Wasserstein metric; the training error is between the training data and the PCA estimate and the test error is between the true distribution and the PCA estimate.

distributions. In other words, generative modeling treats the data itself as a distribution and not as a set of data points. As a consequence, although there may exist more than one interpolating solution $\boldsymbol{\theta}$, there is only one unique interpolating distribution $p_{\boldsymbol{\theta}}$. Importantly, this result is not specific to GANs but to any generative model that is trained to minimize the distance between the generated distribution and a fixed distribution.

We now narrow our focus to a specific data model to help understand the constant regime of generalization errors. Recall from Section 2 that in the LQG setting, PCA is a solution for the optimal linear generator. Hence, we can study PCA solutions and evaluate them using the 2-Wasserstein metric (Equation (1)) to see the generalization error of the linear generator in the LQG setting. We assume that our training data $\{\mathbf{x}_i\}_{i=1}^n$ are realizations of a random vector $\mathbf{x} \in \mathbb{R}^d$ that satisfies the noisy linear model $\mathbf{x} = \boldsymbol{\Gamma}\mathbf{z} + \boldsymbol{\epsilon}$. Here $\boldsymbol{\Gamma} \in \mathbb{R}^{d \times m}$ is a deterministic rank $m$ matrix (for $m < d$), $\mathbf{z} \in \mathbb{R}^m$ is a latent random vector of a zero-mean isotropic Gaussian distribution, and $\boldsymbol{\epsilon} \sim \mathcal{N}(\mathbf{0}, \sigma^2 \mathbf{I}_d)$ is a noise vector. The true latent dimension $m$ is unknown to the learner; hence, we will pick $k > 0$ and learn a generator matrix $\mathbf{G} \in \mathbb{R}^{d \times k}$. The true, noiseless distribution is Gaussian: $\mathbf{x}_{\text{true}} = \boldsymbol{\Gamma}\mathbf{z} \sim \mathcal{N}(\mathbf{0}, \boldsymbol{\Gamma}\boldsymbol{\Gamma}^\top)$. Thus, if the learned latent dimension $k$ equals the true latent dimension $m$, then $\mathbf{G} = \boldsymbol{\Gamma}$ is an optimal solution. We assume that $m < d$, hence the covariance matrix of $\mathbf{x} \sim \mathcal{N}(\mathbf{0}, \boldsymbol{\Gamma}\boldsymbol{\Gamma}^\top + \sigma^2 \mathbf{I}_d)$ is the sum of a low rank covariance matrix $\boldsymbol{\Gamma}\boldsymbol{\Gamma}^\top$ and a full rank noise covariance matrix, our choice of $k$ will affect how much we overfit to the noise distribution.

We consider $m < n < d$, i.e., the number of training examples $n$ is higher than the true latent space dimension $m$, and lower than the data dimension $d$, for several reasons. Most importantly, data is often assumed to lie on a low dimensional manifold in a higher dimensional space. Thus if $m \geq d$, then $\boldsymbol{\Gamma}\boldsymbol{\Gamma}^\top$ will have rank $d$ and the noiseless data $\mathbf{x}_{\text{true}}$ will have a non-zero probability of being in any open set in $\mathbb{R}^d$, which is clearly not true for many types of data, such as natural images. We also choose to study $m < d$ because it will allow our model to overfit (when the learned latent dimension $k > m$). Now we turn to our choice of $n$ and note that if $n \geq d$, we get the typical U-shaped curve of the bias-variance tradeoff for generalization error as a function of the learned latent dimension $k$. If $n \leq m$, then the generalization error is just monotonically decreasing in $k$ and is of little interest because it is unlikely that we overfit to our data. For these reasons, we consider only $m < n < d$, which permits study of the double descent phenomenon.

We train a linear GAN by picking the top $k$ principal components ($k \leq d$), namely, minimizing the training loss

$$\mathcal{L}^{\text{train}}(\mathbf{G}, \mathbf{X}) = \|(\mathbf{I}_d - \mathbf{G}\mathbf{G}^\top)\mathbf{X}\|_F^2 \tag{2}$$

under the constraint that the $d \times k$ matrix $\mathbf{G}$ has orthonormal columns. Moreover, $\mathbf{X} \in \mathbb{R}^{d \times n}$ denotes the data matrix with $n$ training examples as its columns. If $k > n$, we run out of nonzero eigenvalues and cannot add any more; the learned generator interpolates by producing zero training error. However, the test error

will increase if we learn noise, i.e., if the eigenvalues and eigenvectors of $\mathbf{\Gamma}\mathbf{\Gamma}^\top + \sigma^2\mathbf{I}_d$ are corrupted by the noise covariance $\sigma^2\mathbf{I}_d$. Figure 1 shows the train and test errors for the learned model as a function of the learned latent dimension $k$. We obtain generalization behavior in two stages. First, there is a U-shape with a minimum around $k = m$; then, as the solutions start to interpolate in the overparameterized regime of $k > n$, we observe a constant test error.

To relate this back to Observation 1 and Corollary 1, here the training data distribution $p_f$ is $\mathcal{N}(\widehat{\boldsymbol{\mu}}_\mathbf{x}, \widehat{\boldsymbol{\Sigma}}_\mathbf{x})$, where $\widehat{\boldsymbol{\mu}}_\mathbf{x} \in \mathbb{R}^d$ and $\widehat{\boldsymbol{\Sigma}}_\mathbf{x} \in \mathbb{R}^{d \times d}$ are the empirical mean vector and covariance matrix of the training data, respectively. Roughly speaking, we can think of the generator as learning the true distribution with some noise for the first $m$ components and then just learning noise in the subspace orthogonal to the data; technically, we learn wrong directions in the data for small $k$ if the noise variance $\sigma^2$ is very large. In this setting, where the number of training samples $n$ allows us to interpolate, the best that one can do is to try to guess $m$ by using prior knowledge or training multiple models using cross-validation. These solutions are not satisfactory in many scenarios, so we delve deeper into understanding why the test error in the overparameterized regime is constant. Specifically, we study the overparameterized regime to see if it can be modified in a beneficial way.

## 3.2 Double descent: Getting double descent through actual supervision

Since PCA gives us a solution to GANs trained in the LQG setting, we turn to studying overparameterization in PCA to understand overparameterization in GANs. It was shown that PCA (with soft orthonormality constraints) does exhibit double descent if supervision was added to the training (Dar et al., 2020). This is consistent with our understanding, since supervision will cause the learned map to have fewer degrees of freedom and fit the data exactly. The work in (Dar et al., 2020) focuses on learning a linear subspace for the purpose of dimensionality reduction, therefore we extend that idea to work for the purpose of data generation.

For a start, consider an ideal setting where $n_{\text{sup}}$ out of the $n$ training examples are given with their true latent vectors. Namely, the training dataset $\mathcal{D}$ includes $n_{\text{sup}} \in \{0, \ldots, n\}$ supervised examples $\{(\mathbf{x}_i, \mathbf{z}_i)\}_{i=1}^{n_{\text{sup}}}$ and $n_{\text{unsup}} = n - n_{\text{sup}}$ unsupervised examples $\{\mathbf{x}_i\}_{i=n_{\text{sup}}+1}^{n}$. The training data vectors are organized as the columns of the matrices $\mathbf{X}^{\text{sup}} \in \mathbb{R}^{d \times n_{\text{sup}}}$, $\mathbf{Z}^{\text{sup}} \in \mathbb{R}^{m \times n_{\text{sup}}}$, and $\mathbf{X}^{\text{unsup}} \in \mathbb{R}^{d \times n_{\text{unsup}}}$, respectively.

Since in the ideal setting of this subsection we have true samples of the latent vectors $\mathbf{z}$ that correspond to data points $\mathbf{x}$, this means that we know the true latent space dimension $m$. Hence, we can control the parameterization of the learned model by choosing a latent dimension $k \le m$ via subsampling of coordinates in $\mathbf{z}$ (it turns out that subsampling allows us to optimize over a pseudometric; see Appendix A). Namely, for a set of $k \le m$ unique coordinate indices $\mathcal{S} \subset \{1, \ldots, m\}$, we define $\{\mathbf{z}_{i,\mathcal{S}}\}_{i=1}^{n_{\text{sup}}}$ as the corresponding subvectors of the training data $\{\mathbf{z}_i\}_{i=1}^{n_{\text{sup}}}$. The matrix $\mathbf{Z}_{\mathcal{S}}^{\text{sup}} \in \mathbb{R}^{k \times n_{\text{sup}}}$ has the subsampled vectors $\{\mathbf{z}_{i,\mathcal{S}}\}_{i=1}^{n_{\text{sup}}}$ as its columns.

To train our model, we use a PCA loss term from (2) on the unsupervised portion of the data $\mathbf{X}^{\text{unsup}}$ mixed with a supervised loss term on the supervised portion of the data $\mathbf{Z}^{\text{sup}}, \mathbf{X}^{\text{sup}}$:

$$\mathcal{L}^{\text{train}}(\mathbf{G}, \mathcal{D}) = \frac{1}{n_{\text{sup}}}\|\mathbf{G}\mathbf{Z}_{\mathcal{S}}^{\text{sup}} - \mathbf{X}^{\text{sup}}\|_F^2 + \frac{1}{n_{\text{unsup}}}\|(\mathbf{I}_d - \mathbf{G}\mathbf{G}^\top)\mathbf{X}^{\text{unsup}}\|_F^2 \tag{3}$$

for a generator matrix $\mathbf{G} \in \mathbb{R}^{d \times k}$ (that is not explicitly constrained to have orthonormal columns). Unlike the PCA optimization in (2), the optimizations in the current and following subsections do not include any explicit orthonormal constraints on the columns of the learned matrix $\mathbf{G}$. Here, the supervised portion of the data gives us specific information about $\mathbf{\Gamma}$, which we can use to train a better model. This model is trained by minimizing the loss in Equation (3) with gradient descent.

Figure 2 shows that *our model does indeed exhibit double descent*. However, there are a few limitations with the setup which we will now enumerate.

L1 **This setup requires supervised pairs:** We may not have access to the true latent vectors in practice.

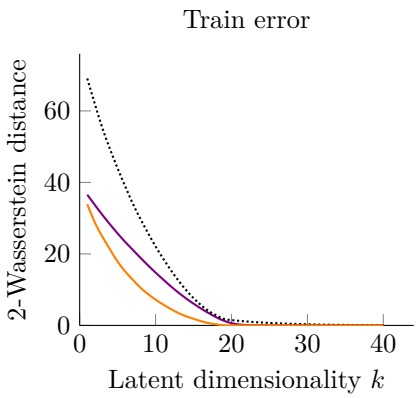 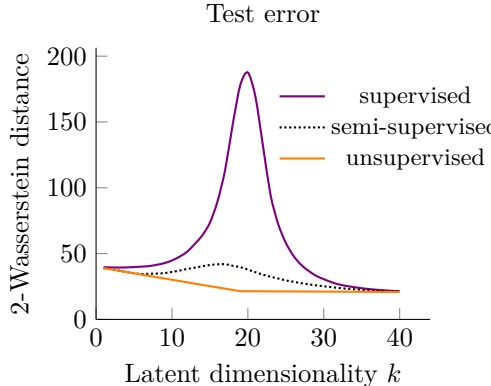

**Figure 2:** The fully supervised model achieves a peak when the latent dimensionality $k$ is equal to the number of training samples $n$. The unsupervised model stops changing as soon as it interpolates at $k = n$. The semi-supervised model with $n_{\text{sup}} = 12$ behaves in a way that is somewhat in-between the other two. The train errors (left subfigure) and test errors (right subfigure) are calculated with the 2-Wasserstein metric; the training error is between the training data and the PCA estimate and the test error is between the true distribution and the PCA estimate. For other values of $n_{\text{sup}}$ and implementation details, see Appendix D.

L2 **This setup requires us to throw away data:** We can only vary $k$ from 1 to $m$ because we are subsampling coordinates from the true latent vectors in $\mathbb{R}^m$. In practice, we would not want to subsample as this throws away potentially useful data information.

L3 **This setup is not typical:** Because $k < m$, we must have $n < m$ to get a peak in the test error at $k = n$ which is not the interesting/typical setting where $m < n < d$ (see Section 3.1). Thus, we are not overfitting, which happens when we learn eigenvectors in the noise directions orthogonal to the data directions, for which $k > m$.

L4 **This setup is supervised, unlike most generative models:** We would like to study generative models, which are typically unsupervised. By adding supervision, we are not actually studying an unsupervised setting but rather a setting which is semi-supervised.

L5 **This setup is not beneficial:** The double descent here actually does not improve performance.

Recall that we want to investigate the overparameterization of generative models to find settings that are realistic and beneficial. We resolve all these problems with pseudo-supervision, defined in the next section.

## 4 Pseudo-supervision: A practical alternative to adding supervision

### 4.1 Definition of pseudo-supervision

Input-output pairs of points are not realistically available in GAN training, which is unsupervised. Therefore, we will *make up latent vectors* that correspond to true data points in our training set. We call these vectors pseudo-supervised latent vectors. Although it may seem odd to partially fabricate training data, there are many advantages to it, starting with not needing access to supervised data (solving L1). Because we have full control over the generation of latent vectors, we can make them of any dimension (solving L2). Consequently, we do not need to know the true latent dimensionality $m$ and can study when $k > m$ (solving L3). Since the pseudo-supervised latent vectors are artificial, this is still an unsupervised problem because we have no supervised data (solving L4). The obvious question is of course "is this beneficial?" The rest of the paper will show that **there are generalization and convergence benefits to pseudo-supervision** (solving L5).

To understand why pseudo-supervision works, consider the supervised scenario discussed in Section 3.2 except with only one supervised sample: $(\mathbf{z}_1, \mathbf{x}_1)$. Now suppose that $\mathbf{z}_{\text{ps}} \in \mathbb{R}^m$ is a completely fabricated

sample, independent of $\mathbf{x}_1$, drawn from the same distribution as $\mathbf{z}_1$. We know that if $\mathbf{G}^{\mathrm{unsup}}$ is a solution to the unsupervised optimization, then so is $\mathbf{G}^{\mathrm{unsup}}\mathbf{U}$ where $\mathbf{U}\mathbf{z}_{\mathrm{ps}} = \mathbf{z}_1$ and $\mathbf{U}$ is an orthonormal matrix (see Theorem 7.3.11 in (Horn & Johnson, 2012)). This is because $(\mathbf{G}^{\mathrm{unsup}}\mathbf{U})(\mathbf{G}^{\mathrm{unsup}}\mathbf{U})^{\top} = \mathbf{G}^{\mathrm{unsup}}(\mathbf{G}^{\mathrm{unsup}})^{\top}$ since positive definite matrices are unique up to a orthonormal transformation. Such a matrix exists if $\|\mathbf{z}_{\mathrm{ps}}\|_2 = \|\mathbf{z}_1\|_2$ because $\mathbf{U}$ is a norm-preserving operator. In other words, it doesn't matter if we use $\mathbf{z}_1$ or $\mathbf{z}_{\mathrm{ps}}$ as long as $\|\mathbf{z}_{\mathrm{ps}}\|_2 = \|\mathbf{z}_1\|_2$. Miraculously, by the curse of dimensionality, $\|\mathbf{z}_{\mathrm{ps}}\|_2 = \|\mathbf{z}_1\|_2$ with high probability if $k$ is large enough! This line of reasoning can be extended past one pseudo-supervised example $n_{\mathrm{ps}} = 1$ to $n_{\mathrm{ps}} = k$ (see Appendix B), after which we incur a penalty for learning a bad representation (because we cannot find an orthonormal matrix which will satisfy the conditions above). Therefore, because of positive definite matrix symmetries and the curse of dimensionality, *we can use pseudo-supervision in a very similar way to supervision without actually knowing any additional information.*

In the following subsections we will define several pseudo-supervised settings, in all of which $n_{\mathrm{ps}}$ out of the $n$ given training examples $\{\mathbf{x}_i\}_{i=1}^n$ are associated with pseudo (i.e., artificial) latent vectors of dimension $k > 0$ (because the true latent dimension is unknown in general). Specifically, the training dataset $\mathcal{D}$ includes $n_{\mathrm{ps}} \in \{0, \ldots, n\}$ pseudo-supervised examples $\{(\mathbf{x}_i, \mathbf{z}_i)\}_{i=1}^{n_{\mathrm{ps}}}$, where $\{\mathbf{z}_i\}_{i=1}^{n_{\mathrm{ps}}}$ are i.i.d. samples of $\mathcal{N}(\mathbf{0}, \mathbf{I}_k)$, and $n_{\mathrm{unsup}} = n - n_{\mathrm{ps}}$ unsupervised examples $\{\mathbf{x}_i\}_{i=n_{\mathrm{ps}}+1}^n$. Therefore, the pseudo-supervised vectors are independent and identically distributed samples from an isotropic Gaussian distribution. The training data vectors are organized as the columns of $\mathbf{X}^{\mathrm{ps}} \in \mathbb{R}^{d \times n_{\mathrm{ps}}}$, $\mathbf{Z}^{\mathrm{ps}} \in \mathbb{R}^{k \times n_{\mathrm{ps}}}$, and $\mathbf{X}^{\mathrm{unsup}} \in \mathbb{R}^{d \times n_{\mathrm{unsup}}}$.

## 4.2 Double descent and superior performance with pseudo-supervision

Our first pseudo-supervised experiment is a straightforward modification of the experiment in Section 3.2. We modify Equation (3) to get the new pseudo-supervised loss

$$\mathcal{L}^{\mathrm{train}}(\mathbf{G}, \mathcal{D}) = \frac{1}{n_{\mathrm{ps}}}\|\mathbf{G}\mathbf{Z}_{\mathcal{S}}^{\mathrm{ps}} - \mathbf{X}^{\mathrm{ps}}\|_F^2 + \frac{1}{n_{\mathrm{unsup}}}\|(\mathbf{I}_d - \mathbf{G}\mathbf{G}^{\top})\mathbf{X}^{\mathrm{unsup}}\|_F^2, \tag{4}$$

where $\mathbf{X}^{\mathrm{ps}} \in \mathbb{R}^{d \times n_{\mathrm{ps}}}$ and $\mathbf{Z}^{\mathrm{ps}} \in \mathbb{R}^{k \times n_{\mathrm{ps}}}$ are the pseudo-supervised matrices. For $n_{\mathrm{unsup}} = 0$ or $n_{\mathrm{ps}} = 0$, we only use the first or second term in the loss, respectively. We provide a detailed explanation of the gradient calculations and optimization procedure in Appendix D. Note that since the pseudo-supervised latent vectors are completely fabricated, we do not have to subsample their coordinates (i.e., as in the supervised setting of Section 3.2) and we can choose $k$ to be any natural number. As shown in the first column of Figure 3, we achieve beneficial double descent behavior of test error. To the best of our knowledge, **this is the first time that double descent has been used beneficially in an unsupervised setting.**

We have extremely low generalization error when $n_{\mathrm{ps}} = n$, even though the loss function does not try to optimize any PCA-type loss. When $n_{\mathrm{ps}} = n$, we have $n_{\mathrm{unsup}} = 0$ and the loss $\mathcal{L}^{\mathrm{train}}(\mathbf{G}, \mathcal{D}) = \|\mathbf{G}\mathbf{Z}_{\mathcal{S}}^{\mathrm{ps}} - \mathbf{X}^{\mathrm{ps}}\|_F^2$ is completely pseudo-supervised. One would expect this scenario to perform poorly since the pseudo-supervised examples do not provide any information, and indeed it does – for small $k$. However, when $k$ is large, we perform well, even though the loss does not attempt to minimize the original PCA loss. Thus, instead of guessing the true latent dimension $m$ that is required for good performance in the standard setting of Section 3.1, **we can simply add pseudo-supervision and increase overparameterization to achieve low generalization error!**

As can be seen from the first column of Figure 3, we achieve better generalization performance via the double descent phenomenon, and we also accelerate training convergence. Interestingly, convergence time actually exhibits double descent as well. The accelerated convergence may be partly due to the unsupervised loss dropping off when $n_{\mathrm{ps}} = n$; however, we will address this in the next section by having a more regularized loss function.

## 4.3 Regularized pseudo-supervision

In the previous section, as $n_{\mathrm{ps}}$ increases, the unsupervised term in the loss drops off. This term, in some sense, regularizes the optimization by encouraging the solution to be orthonormal. This is because, if $\mathbf{G}$ has orthonormal columns, then $(\mathbf{I}_d - \mathbf{G}\mathbf{G}^{\top})\mathbf{x} = 0$ for all $\mathbf{x}$ in the columnspace of $\mathbf{G}$. We will then use the

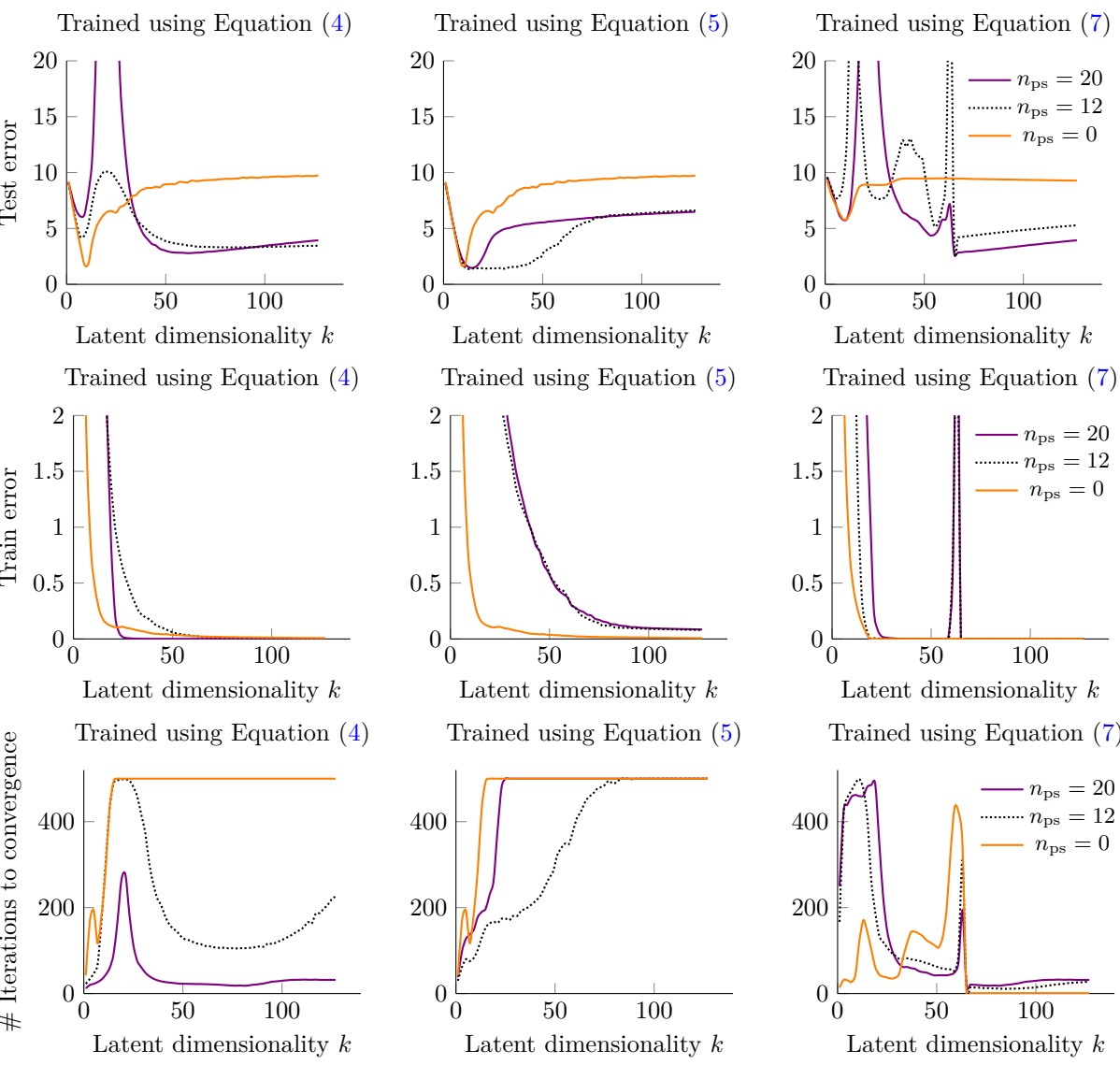

**Figure 3:** Evaluation of test error and training convergence speed in learning of linear GANs using the three different training loss formulations in (4),(5),(7). In the first column of subfigures, we use (4) and get double descent that beats the unsupervised baseline in both generalization performance and convergence speed in the overparameterized range of solutions (the baseline corresponds to the case of no pseudo-supervised training samples $n_{\mathrm{ps}} = 0$). In the second column of subfigures, we use (5) and squash the double descent to get lower generalization error for small latent dimensionality $k$. In the third column of subfigures, we get triple descent (one peak at $k = n$ and one peak at $k = d$) as well as low generalization errors and extremely fast training speed for large $k$. In these experiments, the true data is $m = 10$ dimensional, the data space is $d = 64$ dimensional, and we have $n = 20$ total training data samples. The null estimator ($\mathbf{G} = \mathbf{0}_{d \times k}$) achieves a test error of approximately 13, so all of these models perform better for large enough $k$. For additional plots, see Appendix D.

full data matrix $\mathbf{X}$ (which is a horizontal concatenation of $\mathbf{X}^{\mathrm{ps}}$ and $\mathbf{X}^{\mathrm{unsup}}$) in the second term of the loss function:

$$\mathcal{L}^{\mathrm{train}}(\mathbf{G}, \mathcal{D}) = \frac{1}{n_{\mathrm{ps}}} \|\mathbf{G}\mathbf{Z}_{\mathcal{S}}^{\mathrm{ps}} - \mathbf{X}^{\mathrm{ps}}\|_F^2 + \frac{1}{n} \|(\mathbf{I}_d - \mathbf{G}\mathbf{G}^\top)\mathbf{X}\|_F^2. \tag{5}$$

The results for this optimization are shown in the second column of Figure 3.

This regularized setting with pseudo-supervision outperforms the completely unsupervised setting, but we do not interpolate (i.e., we do not achieve zero train loss), and, consequently, do not see a double descent. This is typical for more regularized problems, as regularization tends to attenuate the double descent phenomenon (see, e.g., for orthonormality constraints in (Dar et al., 2020), or for ridge regularization in (Hastie et al., 2019; Nakkiran et al., 2020)). However, this suggests that the relative importance between the first and second term may significantly impact double descent behavior. More specifically, the only difference between this optimization and the one discussed in Section 4.2 is that the second term uses all the data even when $n_{\text{ps}} > 0$. Thus, we can think of the second term as a regularizer for the loss. On the other hand, we can view the first term as constraining the optimization to fit our pseudo-supervised pairs of points, and thus also a regularizer. Therefore, depending on the point of view, each term can regularize the loss.

Since either of the terms in the training loss in (5) can be perceived as a regularizer, we augment (5) with disproportionate weighting in order to see if this affects the generalization behavior (e.g., the existence of double descent phenomena):

$$\mathcal{L}^{\text{train}}(\mathbf{G}, \mathcal{D}) = \frac{\alpha}{n_{\text{ps}}} \|\mathbf{G}\mathbf{Z}_{\mathcal{S}}^{\text{ps}} - \mathbf{X}^{\text{ps}}\|_F^2 + \frac{1-\alpha}{n} \|(\mathbf{I}_d - \mathbf{G}\mathbf{G}^\top)\mathbf{X}\|_F^2, \tag{6}$$

for $\alpha \in [0, 1]$. A figure of the results is shown in Appendix D. Surprisingly, weighting the loss function in this manner actually does achieve double descent, which leads to lower test error. We discuss this model here in order to highlight that the relative importance between the pseudo-supervised and unsupervised loss terms can induce double descent behavior.

Since we do not exhibit double descent using Equation (5), it is interesting to characterize how the pseudo-supervision term affects the solution set of the problem. The next two theorems characterize the usual unsupervised solutions and the pseudo-supervised solutions. Their proofs are included in Appendix C.

**Theorem 1.** *Suppose that $\mathbf{X} \in \mathbb{R}^{d \times n}$ has full rank of $\min\{d, n\}$. For the unsupervised loss $L_{unsup}^\top(\mathbf{G}, \mathbf{X}) \triangleq \|(\mathbf{I}_d - \mathbf{G}\mathbf{G}^\top)\mathbf{X}\|_F^2$, let $\mathcal{S}_{unsup}^\top(k) \triangleq \{\mathbf{G} \in \mathbb{R}^{d \times k} : L_{unsup}^\top(\mathbf{G}, \mathbf{X}) = 0\}$ be the set of interpolating solutions. Then,*

1. *$\mathcal{S}_{unsup}^\top(k) = \emptyset$ if $n > k$.*

2. *$\mathcal{S}_{unsup}^\top(k)$ is a smooth manifold of dimension $\frac{n(n-1)}{2}$ when $n = k$.*

3. *$\mathcal{S}_{unsup}^\top(k)$ is the union of $\binom{n}{k}$ smooth manifolds of dimension $\frac{n(n-1)}{2}(k-n)(d-n)$ when $k > n$.*

**Theorem 2.** *Suppose that $\mathbf{X} \in \mathbb{R}^{d \times n}$ has full rank of $\min\{d, n\}$ and let $\lambda > 0$ be given. For the pseudo-supervised loss $L_{ps}^\top(\mathbf{G}, \mathbf{X}; \lambda) \triangleq \frac{\lambda}{n_{ps}} \|\mathbf{G}\mathbf{Z}_{\mathcal{S}}^{ps} - \mathbf{X}^{ps}\|_F^2 + \frac{1}{n} \|(\mathbf{I}_d - \mathbf{G}\mathbf{G}^\top)\mathbf{X}\|_F^2$, let $\mathcal{S}_{ps}^\top(k) \triangleq \{\mathbf{G} \in \mathbb{R}^{d \times k} : L_{ps}^\top(\mathbf{G}, \mathbf{X}, \lambda) = 0\}$ be the set of interpolating solutions. Then,*

1. *$\mathcal{S}_{ps}^\top(k) = \emptyset$ if $n > k$ and $\mathbf{Z} \in \mathbb{R}^{k \times n}$ is arbitrary.*

2. *$\mathcal{S}_{ps}^\top(k)$ has only one element if $n = k$ and $\mathbf{Z} \in \mathbb{R}^{k \times n}$ is given so that $\mathbf{Z}^\top\mathbf{Z} = \mathbf{X}^\top\mathbf{X}$.*

3. *$\mathcal{S}_{ps}^\top(k)$ is the union of $\binom{n}{k}$ smooth manifolds of dimension $(k-n)(d-n)$ if $k > n$ and $\mathbf{Z} = \begin{bmatrix} \mathbf{Z}_1 \\ \mathbf{0} \end{bmatrix} \in \mathbb{R}^{k \times n}$ is given so that $\mathbf{Z}_1^\top\mathbf{Z}_1 = \mathbf{X}^\top\mathbf{X}$.*

Note that although there exists many unsupervised solutions (Theorem 1), the pseudo-supervised solutions (Theorem 2) depend heavily on the condition that $\mathbf{Z}^\top\mathbf{Z} = \mathbf{X}^\top\mathbf{X}$. Indeed this condition does not happen in practice with a Gaussian $\mathbf{Z}$, resulting in no interpolation and a regularizing effect. This restrictive condition comes from the transpose in the unsupervised term. In the next section we will relax this into a pseudo-inverse in order to interpolate better.

### 4.4 Triple descent and huge latent spaces

The similar losses of Equations ([4](#)) to ([6](#)) indirectly encourage learning semi-orthogonal generator matrices. We can relax this constraint and let our generator learn more complex linear functions by optimizing

$$\mathcal{L}^{\text{train}}(\mathbf{G}, \mathcal{D}) = \frac{1}{n_{\text{ps}}} \|\mathbf{G}\mathbf{Z}_{\mathcal{S}}^{\text{ps}} - \mathbf{X}^{\text{ps}}\|_F^2 + \frac{1}{n} \|(\mathbf{I}_d - \mathbf{G}\mathbf{G}^\dagger)\mathbf{X}\|_F^2, \tag{7}$$

where $\mathbf{G}^\dagger$ is the Moore-Penrose pseudo-inverse of the matrix $\mathbf{G}$. Training this loss may seem similar to the others, but the results are quite different.

With this new loss, we achieve triple descent and desirable generalization and convergence behavior when the latent dimensionality $k$ is larger than the data space dimensionality $d$ (third column of Figure [3](#)). This scenario is most closely related to neural networks because the models that we learn are very general and typically not constrained (e.g., to have orthonormal layers). Moreover, the pseudo-supervised optimization converges to a solution which beats the unsupervised baseline with few iterations.

Just as we did in Section [4.4](#), we will characterize the solution sets for unsupervised pseudoinverse loss and the pseudo-supervised pseudoinverse loss. The next two theorems do this and show that we no longer have the restrictive $\mathbf{Z}^\top\mathbf{Z} = \mathbf{X}^\top\mathbf{X}$ condition. Their proofs are located in Appendix [C](#).

**Theorem 3.** *Suppose that $\mathbf{X} \in \mathbb{R}^{d \times n}$ has full rank of $\min\{d, n\}$. For the unsupervised loss $L_{unsup}^\dagger(\mathbf{G}, \mathbf{X}) \triangleq \|(\mathbf{I}_d - \mathbf{G}\mathbf{G}^\dagger)\mathbf{X}\|_F^2$, let $\mathcal{S}_{unsup}^\dagger(k) \triangleq \{\mathbf{G} \in \mathbb{R}^{d \times k} : L_{unsup}^\dagger(\mathbf{G}, \mathbf{X}) = 0\}$ be the set of interpolating solutions. Then,*

1. *$\mathcal{S}_{unsup}^\dagger(k) = \emptyset$ if $n > k$.*

2. *$\mathcal{S}_{unsup}^\dagger(k)$ is a smooth manifold of dimension $n^2$ when $n = k$.*

3. *$\mathcal{S}_{unsup}^\dagger(k)$ is the union of $\binom{n}{k}$ smooth manifolds of dimension $n^2(k-n)d$ when $k > n$.*

**Theorem 4.** *Suppose that $\mathbf{X} \in \mathbb{R}^{d \times n}$ has full rank of $\min\{d, n\}$ and let $\lambda > 0$ be given. $L_{ps}^\dagger(\mathbf{G}, \mathbf{X}; \lambda) \triangleq \frac{\lambda}{n_{ps}}\|\mathbf{G}\mathbf{Z}_{\mathcal{S}}^{ps} - \mathbf{X}^{ps}\|_F^2 + \frac{1}{n}\|(\mathbf{I}_d - \mathbf{G}\mathbf{G}^\dagger)\mathbf{X}\|_F^2$, let $\mathcal{S}_{ps}^\dagger(k) \triangleq \{\mathbf{G} \in \mathbb{R}^{d \times k} : L_{ps}^\dagger(\mathbf{G}, \mathbf{X}, \lambda) = 0\}$ be the set of interpolating solutions. Then,*

1. *$\mathcal{S}_{ps}^\dagger(k) = \emptyset$ if $n > k$.*

2. *$\mathcal{S}_{ps}^\dagger(k)$ has only one element when $n = k$.*

3. *$\mathcal{S}_{ps}^\dagger(k)$ an affine space of dimension $(k-n)d$ when $k > n$.*

## 5 Nonlinear GANs: Double descent and faster training

In this section we show that double descent can occur in nonlinear, multilayer GANs trained with pseudo-supervision. Finding the right experimental setting for double descent was difficult because the level of parameterization is much harder to quantify in a multilayer network. We still determined the overparameterization solely by modifying the latent dimensionality $k$ and not by making the networks wider or deeper. The right side of Figure [4](#) shows double descent for our pseudo-supervised model. We trained a total of 430 GANs (with different latent dimensionalities and initializations) to make that figure, which is why a study like this would be computationally prohibitive on models that take a significant amount of time to train.

We also found that these realistic GANs trained with pseudo-supervision converge to a good solution much faster than they would have without the pseudo-supervision. Figures [4](#) to [7](#) show the test errors as training progressed for different latent dimensionalities. The pseudo-supervised models converge much faster and

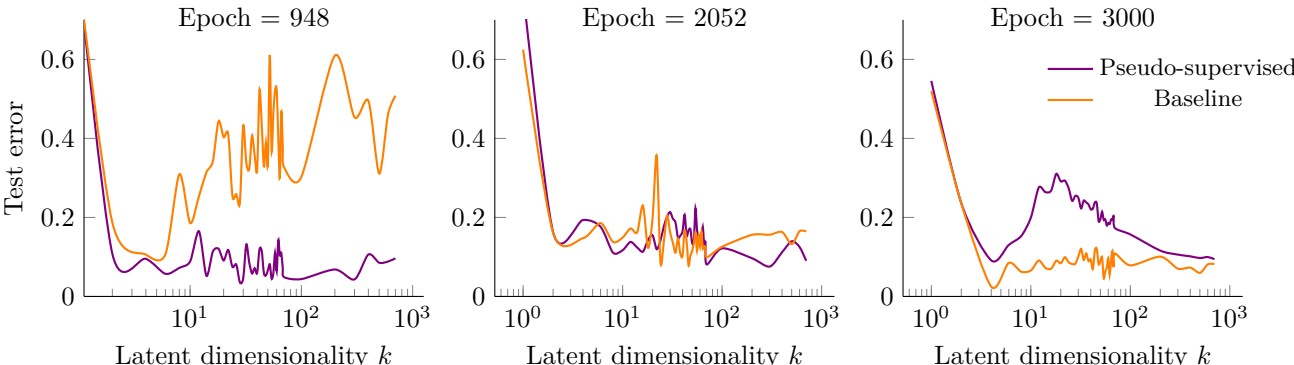

**Figure 4:** Test errors for multilayer, nonlinear GANs trained on the MNIST digit dataset. On the left we see that the baseline error resembles a noisy version of the test error in Figure 1, characterized by an initial dip and then high levels of error. Our pseudo-supervision training beats the baseline here. As we continue to train (epoch 2052), we see that the baseline error reduces, which may be due to some kind of implicit regularization. On the right, our pseudo-supervised model achieves double descent at epoch 3000. Here the test error is measured by geometry score.

performed very well for models trained on both MNIST and CelebA. On the MNIST training, the pseudo-supervised models converged to the lowest test error after only about 750 epochs compared to about 1,500 epochs in the baseline case. On the CelebA training, the pseudo-supervised models converged to the lowest test error after only about 10,000 epochs compared to about 40,000 epochs in the baseline case.

The test error in Figure 4 for the MNIST baseline had an initial dip then continued up to high levels around epoch 948, suggesting overfitting similar to what we saw in the linear models. We suspect that this overfitting was reduced as we continued to train because of some internal regularization, such as the batch norm in the model.

We performed these experiments with some non-standard procedures to aid in our understanding of generalization and double descent phenomena in GANs. In this work, we are not concerned with training state-of-the-art GANs. For this reason, our experiments are on MNIST (LeCun et al., 1998) and CelebA (Liu et al., 2015). Since MNIST is not very complex, we only use a random subset of 4,096 training data points and perform gradient descent using a gradient penalized Wasserstein GAN[1] (for SGD results, see Appendix E). Commonly used performance metrics such as FID (Heusel et al., 2017) and IS (Salimans et al., 2016) are made for natural images since they use the Inception v3 (Szegedy et al., 2016) model trained on ILSVRC 2012 (Russakovsky et al., 2015). Therefore, we use the geometry score (Khrulkov & Oseledets, 2018), which is better suited for MNIST[2]. The experiments on CelebA only uses a random subset of 128 training data points for the same reasons. However, we use FID to evaluate the CelebA experiments with FID evaluated using 128 images for computational efficiency. In addition to the reasons stated above, for the MNIST and CelebA experiments, we use subsets of data because if we use the whole dataset we run out of orthogonal pseudo-supervised vectors. This is a limitation of our work when it comes to smaller networks which have small latent spaces compared to the number of data points. See Appendix E for more details on the training.

## 6    Conclusion

We have demonstrated that pseudo-supervision can be used to achieve beneficial double descent phenomena in unsupervised models, specifically in linear GANs and nonlinear, multilayer GANs. Pseudo-supervision can help accelerate training and lower generalization error. This opens up areas of research in understanding overparameterization and double descent behavior in unsupervised models. Moreover, our findings suggest that an empirical study on ImageNet with more complex networks is beneficial to improve state-of-the-art generalization error and convergence speed.

---

[1]The architecture implementation can be found here
[2]The geometry score implementation can be found here

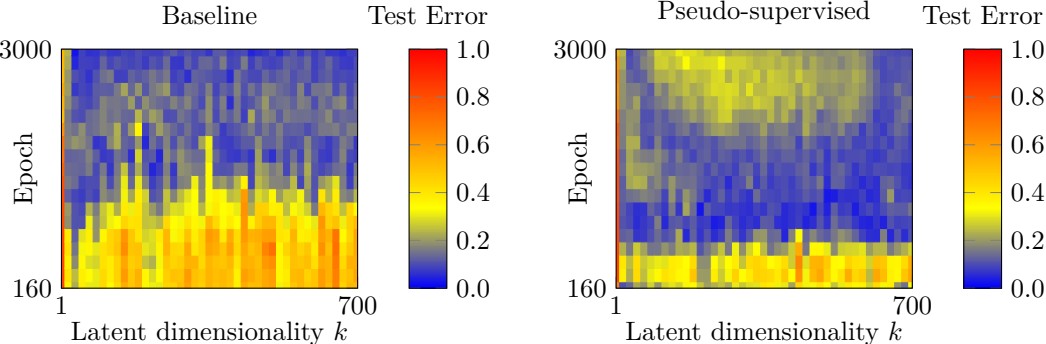

**Figure 5:** These test error heatmaps for multilayer, nonlinear GANs trained on MNIST show that the pseudo-supervised models converge faster than the baseline models. The baseline model has high test error until around epoch 1500, unlike the pseudo-supervised models which have the test error drop off at around epoch 750. The baseline model only beats the pseudo-supervised model later in the training (around epoch 2500), when the pseudo-supervised loss increases and admits a double descent shape. The test error is measured by geometry score here. The $k$-axis is plotted so that each column corresponds to the next entry for better visualization, even though the spacing is $k \in \{1, 2, 4, 6, \dots, 70, 100, 200, 300, \dots, 700\}$.

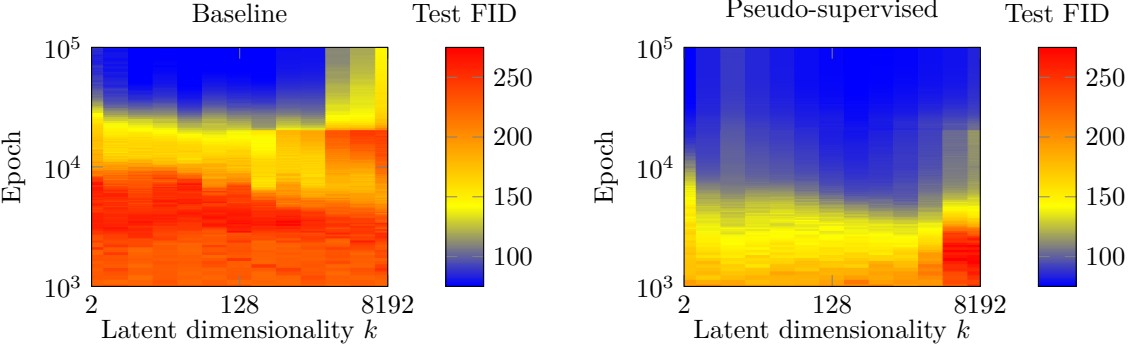

**Figure 6:** These test error (measured via FID) heatmaps for multilayer, nonlinear GANs trained on CelebA show that the pseudo-supervised models converge faster than the baseline models. The baseline model has high test error until around epoch 40,000, unlike the pseudo-supervised models which have the test error drop off at around epoch 10,000. The baseline model only beats the pseudo-supervised model later in the training, however only in the lower parameterized regime. The $k$-axis is plotted for $k \in \{2, 4, 8, \dots, 4096, 8192\}$ and the epoch axis is plotted from 1000 to 100,000.

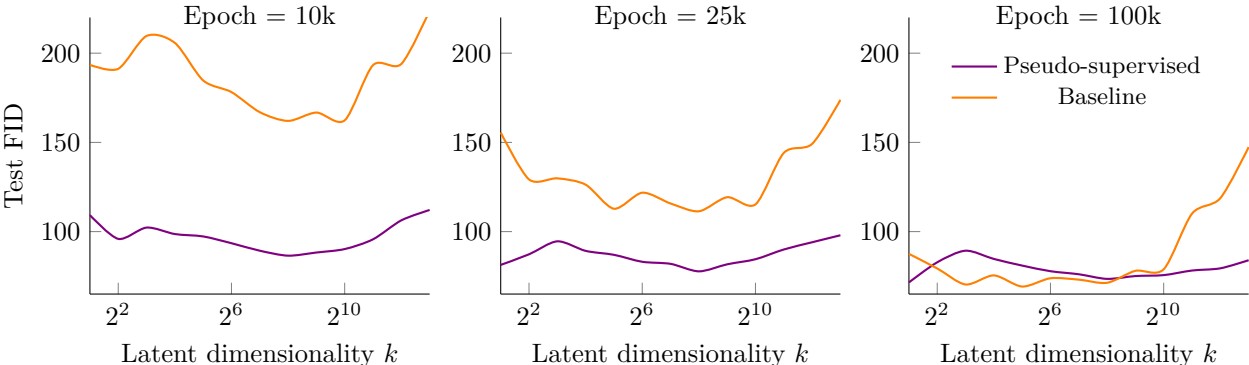

**Figure 7:** Test errors (measured via FID) for a multilayer, nonlinear GAN trained on the CelebA dataset. On the left we see that the baseline error is quite high and our pseudo-supervision training has almost converged after only 10k epochs. As we continue to train (epoch 25k), we see that the baseline error reduces along with the pseudo-supervised error. On the right, we see that although the baseline error can even beat the pseudo-supervised error for certain model parameterizations, this is not the case for highly overparameterized models where pseudo-supervision still outperforms the baseline.

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

## Appendices

The appendices below support the main paper as follows. Appendix A provides additional details on how subsampling (or zeroing) coordinates of the data is equivalent to training with a pseudometric as discussed in Section 3.2 of the main paper. In Appendix B we expand on pseudo-supervision and explain when it can be used to mimic supervision. Appendix D includes additional empirical results and details for the linear GAN problems from Sections 3.2 and 4.2 to 4.4 of the main paper. Appendix E provides additional experimental results and details for the multilayer, nonlinear GAN from Section 5 of the main paper; we also include results for SGD-based training of GANs using the complete MNIST dataset.

## A    Training with pseudometric and subsampling data

The problem in Section 3.1 is that we are optimizing over a metric $q$, which has a definiteness property, i.e., $q(x, y) = 0$ if and only if $x = y$. If we relax this property, we are left with a pseudometric; similarly, we can relax this property to obtain a non-definite $f$-divergence. Interestingly, we found that subsampling coordinates of the data is equivalent to using a pseudometric, and we will use subsampling in the next section to control our level of parameterization. We provide a detailed discussion on the pseudometric formulation in Appendix A since several papers on double descent use feature subsampling to control the parameterization of the model (Belkin et al., 2019b; Dar et al., 2020; Dar & Baraniuk, 2022).

**Proposition 1.** *Let $q_d$ be any metric on $\mathbb{R}^d$ and $q_k$ be any metric on $\mathbb{R}^k$. Suppose that $\mathbf{x}_d, \mathbf{y}_d \in \mathbb{R}^d$ are subsampled (in the same way) to $\mathbf{x}_k, \mathbf{y}_k \in \mathbb{R}^k$, with $k < d$, i.e., the elements of $\mathbf{x}_k$ and $\mathbf{y}_k$ are a subset of the elements of $\mathbf{x}_d$ and $\mathbf{y}_d$, respectively. Then, the function $q' : \mathbb{R}^d \times \mathbb{R}^d \to \mathbb{R}$ defined as $q'(\mathbf{x}_d, \mathbf{y}_d) = q_k(\mathbf{x}_k, \mathbf{y}_k)$ is a pseudo-metric in $\mathbb{R}^d$.*

*Proof.* We first show that $q'$ is strictly semi-definite. For all $\mathbf{x}_d \in \mathbb{R}^d$ we have that $q'(\mathbf{x}_d, \mathbf{x}_d) = q_k(\mathbf{x}_k, \mathbf{x}_k) = 0$. Note however, that if we modify an element of $\mathbf{x}_d$ which is not present in $\mathbf{x}_k$ to create $\mathbf{x}'_d \neq \mathbf{x}_d$ we will still get $q'(\mathbf{x}_d, \mathbf{x}'_d) = q_k(\mathbf{x}_k, \mathbf{x}_k) = 0$, implying that $q'$ is not strictly definite and hence that $q'$ cannot be a proper metric. Moreover, we see that $q'$ is symmetric because for all $\mathbf{x}_d, \mathbf{y}_d \in \mathbb{R}^d$ we have that $q'(\mathbf{x}_d, \mathbf{y}_d) = q_k(\mathbf{x}_k, \mathbf{y}_k) = q_k(\mathbf{y}_k, \mathbf{x}_k) = q'(\mathbf{y}_d, \mathbf{x}_d)$.

Now we must show that the triangle inequality holds for $q'$. Suppose that $\mathbf{x}_d, \mathbf{y}_d, \mathbf{z}_d \in \mathbb{R}^d$ are subsampled to $\mathbf{x}_k, \mathbf{y}_k, \mathbf{z}_k \in \mathbb{R}^k$. Then,

$$
\begin{aligned}
q'(\mathbf{x}_d, \mathbf{z}_d) &= q_k(\mathbf{x}_k, \mathbf{z}_k) \\
&\leq q_k(\mathbf{x}_k, \mathbf{y}_k) + q_k(\mathbf{y}_k, \mathbf{z}_k) && \text{(triangle inequality on } q_k) \\
&= q'(\mathbf{x}_d, \mathbf{y}_d) + q'(\mathbf{y}_d, \mathbf{z}_d),
\end{aligned}
$$

as desired. Thus, $q'$ is a pseudo-metric. $\qquad\square$

### A.1    Subsampling the data features

In Section 3.1 we saw that if we optimize an objective function which is a metric (Rudin, 1964) or an $f$-divergence (Rényi et al., 1961; Nowozin et al., 2016), the resulting generalization error will be constant for any interpolating solution. This is due to the definiteness of the metric or $f$-divergence. In this section we will relax this property for the 2-Wasserstein metric (Villani, 2003; 2008); extensions to this relaxation can be done for $f$-divergences and other metrics. The resulting mathematical object is called a pseudometric (Royden & Fitzpatrick, 1988), which has been studied thoroughly in the context of $L^p$ metrics in Banach spaces (Axler, 2020; Royden & Fitzpatrick, 1988).

**Definition 1.** *We denote $q_d$ to be the standard Euclidean metric on $\mathbb{R}^d$ (Rudin, 1964). Let $P(\mathbb{R}^d)$ be the set of all probability distributions defined on the measurable space $(\mathbb{R}^d, \mathcal{B}(\mathbb{R}^d))$, where $\mathcal{B}(\mathbb{R}^d)$ is the Borel $\sigma$-algebra on $\mathbb{R}^d$ (Axler, 2020; Rudin, 1987). We denote $\mathcal{W}_d : P(\mathbb{R}^d) \times P(\mathbb{R}^d) \to \mathbb{R}$ to be the 2-Wasserstein metric:*

$$
\mathcal{W}_d(P, P') = \sqrt{\inf_{\gamma \in \Pi(P, P')} \int_{\mathbb{R}^d \times \mathbb{R}^d} q_d^2(\mathbf{x}, \mathbf{y}) d\gamma(\mathbf{x}, \mathbf{y})},
$$

*where $\gamma \in \Pi(P, P')$ is any joint distribution of $P$ and $P'$. For a set $A \subset \{1, \ldots, d\}$, we define the pseudometric $\mathcal{W}_{d,A} : \mathbb{R}^d \times \mathbb{R}^d \to \mathbb{R}$ to be the 2-Wasserstein metric on $\mathbb{R}^{d-|A|}$ on the indices not in $A$. For example, if $P_{2,\ldots,d}, P'_{2,\ldots,d}$ are the marginals (after integrating out the first component) of $P$ and $P'$, respectively, then*

$$\mathcal{W}_{d,\{1\}}(P, P') := \mathcal{W}_{d-1}(P_{2,\ldots,d}, P'_{2,\ldots,d}).$$

*Clearly, $\mathcal{W}_{d,A}$ is a pseudometric as it derives all metric properties from $\mathcal{W}_{d-|A|}$ except the definiteness property.*

This pseudometric is constructed by integrating out certain coordinates of the distributions and using a metric on the resulting marginal distributions. Therefore it is possible to have zero distance between two distributions that differ along the coordinates which are integrated out. This is equivalent to subsampling or zeroing out the desired coordinates, which we will shortly show. Thus, for the linear case, we can learn a generator $\mathbf{G}$ which maps our latent space to $\mathbb{R}^d$ and which learns the training data distributions $p_f$ except for the ignored coordinates. Of course, now we have a whole (affine) subspace of matrices $\mathbf{G} \in \mathbb{R}^{d \times k}$ that we can learn. In other words, using a pseudometric, an interpolating solution $\mathbf{G} \in \mathbb{R}^{d \times k}$ forms an affine subspace of $\mathbb{R}^{d \times k}$ if modified along the ignored coordinates. As we will see in Appendix B, we can also transform $\mathbf{G}$ by an orthonormal transformation to get more degrees of freedom than just this affine space. In this setting, the min-norm solution will not project anything on the ignored coordinates.

**Theorem 5.** *Let $P$ and $P'$ be two distributions defined on $\mathbb{R}^d$. Let $A \subset \{1, \ldots, d\}$ be a subset of the axis indices. We define a new distribution $Q_A$ on $\mathbb{R}^d$ as the product of $|A|$ univariate point masses at $0$ and the marginal distribution $P_{A^C}$. The point masses are located so that the univariate marginals of $Q_A$ are point masses along the coordinates in $A$. We define $Q'_A$ similarly. Then,*

$$\mathcal{W}_{d,A}(P, P') := \mathcal{W}_{d-|A|}(P_{A^C}, P'_{A^C}) = \mathcal{W}_d(Q_A, Q'_A). \tag{8}$$

*Proof.* An application of Tonelli's Theorem (Axler, 2020) shows that

$$
\begin{aligned}
\mathcal{W}_d(Q_A, Q'_A) &= \sqrt{\inf_\gamma \int_{\mathbb{R}^d \times \mathbb{R}^d} q_d^2(\mathbf{x}, \mathbf{y}) d\gamma} \\
&= \sqrt{\inf_\gamma \sum_{i=1}^d \int_{\mathbb{R}^d \times \mathbb{R}^d} |x_i - y_i|^2 d\gamma} \\
&= \sqrt{\inf_\gamma \sum_{i \in A} \int_{\mathbb{R}^d \times \mathbb{R}^d} |x_i - y_i|^2 d\gamma + \sum_{i \in A^C} \int_{\mathbb{R}^d \times \mathbb{R}^d} |x_i - y_i|^2 d\gamma} \\
&= \sqrt{\inf_\gamma \int_{\mathbb{R}^{2|A|}} \sum_{i \in A} |x_i - y_i|^2 d\gamma_A + \int_{\mathbb{R}^{2(d-|A|)}} \sum_{i \in A^C} |x_i - y_i|^2 d\gamma_{A^C}} \quad \text{(Tonelli)} \\
&= \sqrt{\inf_{\gamma_{A^C}} \int_{\mathbb{R}^{2(d-|A|)}} \sum_{i \in A^C} |x_i - y_i|^2 d\gamma_{A^C}} \quad (*) \\
&= \mathcal{W}_{d-|A|}(P_{A^C}, P'_{A^C}) \\
&= \mathcal{W}_{d,A}(P, P'),
\end{aligned}
$$

where $\gamma_A$ and $\gamma_{A^C}$ are the joints of the marginals over $A$ and over $A^C$, respectively. We also use independence when using Tonelli's Theorem, because $Q_A$ and $Q'_A$ are product measures by construction. In $(*)$, we pick $\gamma_A$ to be the independent joint distribution so that each random variable with index in $A$ is independent. Since each of these random variables is identical, the integral term on the left vanishes and is therefore the minimizer of the infimum. $\square$

Theorem 5 shows that we can train with a pseudometric by simply zeroing the coordinates of the data that we wish to ignore; alternatively, we can also subsample the features so that we keep the features with indices

in $\mathcal{A}^C$. This allows us to consider a pseudometric $\mathcal{W}_{d,A}$ which is invariant to the data features with indices in $A$. Suppose that we instead want $\mathcal{W}_{d,A}$ to be invariant to a specific subspace. It turns out that these two concepts are closely related.

**Theorem 6.** *Let $V \subset \mathbb{R}^d$ be a subspace spanned by the orthonormal vectors $\mathbf{v}_1, \ldots, \mathbf{v}_m$; the rest of $\mathbb{R}^d$ is spanned by $\mathbf{v}_{m+1}, \ldots, \mathbf{v}_d$ so that $\{\mathbf{v}_i\}_{i=1}^d$ is an orthonormal basis for $\mathbb{R}^d$. We also have a data matrix $\mathbf{X} \in \mathbb{R}^{d \times n}$. Then, we can construct a pseudometric $\mathcal{W}_{d,V}$ to be invariant to the subspace $V$ by replacing the first $m$ rows of $\mathbf{U}^\top \mathbf{X}$ with zeros for $\mathbf{U} = \begin{bmatrix} \mathbf{v}_1 & \ldots & \mathbf{v}_d \end{bmatrix} \in \mathbb{R}^{d \times d}$.*

*Proof.* Let $\mathbf{v} \in V$ be given. Then, we can write $\mathbf{v} = \sum_{i=1}^m c_i \mathbf{v}_i$. Clearly, we have that $\mathbf{U}^\top \mathbf{v} = \sum_{i=1}^m c_i \mathbf{U}^\top \mathbf{v}_i = \begin{bmatrix} c_1 & \ldots & c_m & 0 & \ldots & 0 \end{bmatrix}^\top$. Similarly, if $\mathbf{w} \in V$ is arbitrary, then we have that $\mathbf{U}^\top \mathbf{w} = \begin{bmatrix} a_1 & \ldots & a_m & a_{m+1} & \ldots & a_d \end{bmatrix}^\top$ for some numbers $a_i \in \mathbb{R}$. Hence, by replacing the first $m$ coordinates by 0 we project onto the subspace orthogonal to $V$. Applying $\mathbf{U}^\top$ to each column of $\mathbf{X}$ is equivalent to computing $\mathbf{U}^\top \mathbf{X}$. $\qquad\square$

Thus, without loss of generality, we consider only subsampling feature indices. If we want to ignore a subspace, we simply multiply our data matrix by the correct matrix $\mathbf{U}$.

### A.2 Subsampling the latent vector coordinates

In the previous section, we considered subsampling the data features. However, we know that supervision has enabled double descent in PCA-type problems (Dar et al., 2020). Thus, we would like to study supervision in the GAN context, as discussed in Section 3.2. In a supervised linear regression setting using the 2-norm loss, we know that we must take a pseudoinverse of the input matrix (Hastie et al., 2009), which induces double descent. In this setting, that is the latent space matrix $\mathbf{Z}$. Therefore, we enable double descent by subsampling the latent vector coordinates. Doing this is very similar to subsampling the features in the data space. For example, if we zero out the first coordinate of the latent distribution, we are essentially zeroing out the subspace corresponding to the first column of the matrix $\mathbf{G}$. Since we learn $\mathbf{G}$, this is a type of adaptive pseudometric procedure, where we learn which subspaces to use and which subspaces to ignore.

## B Pseudo-supervision and the curse of dimensionality

This appendix provides further detail regarding the scenario described in Section 4.1. Suppose that $\mathbf{G} \in \mathbb{R}^{d \times m}$ is a solution which provides zero test error. Now, let $\mathbf{z} \in \mathbb{R}^m$ correspond to the true vector which generates $\mathbf{x} \in \mathbb{R}^d$ so that $\mathbf{G}\mathbf{z} = \mathbf{x}$. Now suppose that $\mathbf{z}_{\mathrm{ps}} \in \mathbb{R}^m$ is any vector so that $\|\mathbf{z}_{\mathrm{ps}}\|_2 = \|\mathbf{z}\|_2$. Then, we can find an orthonormal matrix $\mathbf{U} \in \mathbb{R}^{m \times m}$ so that $\mathbf{U}\mathbf{z}_{\mathrm{ps}} = \mathbf{z}$. We see that $\mathbf{G}\mathbf{U}$ is also a solution which gives zero train error, because the isotropic covariance matrix of the generated distribution is not changed if we right multiply $\mathbf{G}$ with an orthonormal matrix (Horn & Johnson, 2012). However, if we pick $\mathbf{z}_{\mathrm{ps}}$ from $\mathcal{N}(0, \mathbf{I}_m)$ where $m$ is large, we see that $\|\mathbf{z}_{\mathrm{ps}}\|_2 = \|\mathbf{z}\|_2$ with high probability because high dimensional Gaussians concentrate on a thin shell in high-dimensional space (Bishop, 2006). This is typically considered a bad thing, hence its name: the curse of dimensionality. However, here we use the curse of dimensionality to allow fabricated latent vectors $\mathbf{z}_{\mathrm{ps}}$ to mimic supervised latent vectors $\mathbf{z}$. Moreover, we can come up with linearly independent pseudo-supervised latent vectors up to $m$ times, after which we can no longer find an $m \times m$ orthonormal matrix $\mathbf{U}$. The more pseudo-supervised samples we have, the fewer matrices $\mathbf{G}$ we can learn, resulting in faster gradient descent convergence since the feasible set is smaller.

We will encounter a problem if $k < m$, i.e., if the latent dimension we pick is lower than the true latent dimension, because we cannot learn a perfect representation (assuming that the linear operator $\mathbf{\Gamma}$ in the data model is full rank). However, if we let $k$ be larger than $m$, then we can learn a solution which gives us zero test error. Although the true vectors are $m$-dimensional, we can always learn a generator matrix $\mathbf{G}$ which ignores certain coordinates. For such solutions, we can also construct pseudo-supervised samples up to $k$ times. Therefore the overparameterized regime, where $k$ is large, is very desirable from the pseudo-supervised point of view.

If we fix $n_{\mathrm{ps}}$ to some value, note that by the above argument, we will incur a penalty if $k < n_{\mathrm{ps}}$ because we will not be able to find a suitable $\mathbf{U}$. However, if $k$ is larger than $m$ and larger than $n_{\mathrm{ps}}$, we can mimic the behavior of supervised samples because we will be able to find an orthonormal matrix which will transform those pseudo-supervised latent vectors into vectors that equal the true vectors along $m$ coordinates. For this reason, we consider pseudo-supervision when $k$ is large.

## C  The dimension of the solution sets

### C.1  The unsupervised solutions

**Theorem 1.** *Suppose that* $\mathbf{X} \in \mathbb{R}^{d \times n}$ *has full rank of* $\min\{d, n\}$. *For the unsupervised loss* $L_{unsup}^{\top}(\mathbf{G}, \mathbf{X}) \stackrel{\Delta}{=} \|(\mathbf{I}_d - \mathbf{G}\mathbf{G}^{\top})\mathbf{X}\|_F^2$, *let* $\mathcal{S}_{unsup}^{\top}(k) \stackrel{\Delta}{=} \{\mathbf{G} \in \mathbb{R}^{d \times k} : L_{unsup}^{\top}(\mathbf{G}, \mathbf{X}) = 0\}$ *be the set of interpolating solutions. Then,*

1. $\mathcal{S}_{unsup}^{\top}(k) = \emptyset$ *if* $n > k$.

2. $\mathcal{S}_{unsup}^{\top}(k)$ *is a smooth manifold of dimension* $\frac{n(n-1)}{2}$ *when* $n = k$.

3. $\mathcal{S}_{unsup}^{\top}(k)$ *is the union of* $\binom{n}{k}$ *smooth manifolds of dimension* $\frac{n(n-1)}{2}(k-n)(d-n)$ *when* $k > n$.

*Proof for Theorem 1.* Suppose that $n > k$ and, for the sake of a contradiction, that $\mathcal{S}_{\mathrm{unsup}}^{\top}(k) \neq \emptyset$. This means that there exists a $\mathbf{G} \in \mathcal{S}_{\mathrm{unsup}}^{\top}(k)$ so that $\|(\mathbf{I}_d - \mathbf{G}\mathbf{G}^{\top})\mathbf{X}\|_F^2 = 0$. Thus, for each column $\mathbf{x}_i$ of $\mathbf{X}$ we have that $\|(\mathbf{I}_d - \mathbf{G}\mathbf{G}^{\top})\mathbf{x}_i\|_2^2 = 0$. In other words, we know that $\mathbf{G}\mathbf{G}^{\top}\mathbf{x}_i = \mathbf{x}_i$ implies that $\mathbf{G}\mathbf{G}^{\top}$ has at least $n$ eigenvalues of 1 corresponding to $n$ eigenvectors since the samples are linearly independent. By the spectral theorem (Theorem 2.5.6 in Horn & Johnson (2012)), $\mathbf{G}\mathbf{G}^{\top}$ is diagonalizable and thus $\mathrm{rank}(\mathbf{G}\mathbf{G}^{\top}) \geq n$. However, we know that $\mathrm{rank}(\mathbf{G}\mathbf{G}^{\top}) \leq k$ by simple rank inequalities (0.4.5 (a) in Horn & Johnson (2012)), a contradiction. Therefore, we know that $\mathcal{S}_{\mathrm{unsup}}^{\top}(k) = \emptyset$.

Now suppose that $n = k$. We take a singular value decomposition of $\mathbf{X}$ into $\mathbf{X} = \mathbf{U}\mathbf{S}\mathbf{V}^{\top}$, where $\mathbf{U} \in \mathbb{R}^{d \times d}, \mathbf{V} \in \mathbb{R}^{k \times k}$ are real orthogonal matrices and $\mathbf{S} \in \mathbb{R}^{d \times k}$ is zero except on the diagonal (Corollary 2.6.7 in Horn & Johnson (2012)). Then, we let $\mathbf{G}_0 = \mathbf{U}_k$ be the first $k$ columns of $\mathbf{U}$. We see that

$$\mathbf{G}_0 \mathbf{G}_0^{\top} \mathbf{X} = \mathbf{U}_k \mathbf{U}_k^{\top} \mathbf{U} \mathbf{S} \mathbf{V}^{\top} = \mathbf{U}_k \begin{bmatrix} \mathbf{I}_k & \mathbf{0}_{k \times d-k} \end{bmatrix} \mathbf{S} \mathbf{V}^{\top} = \begin{bmatrix} \mathbf{U}_k & \mathbf{0}_{k \times d-k} \end{bmatrix} \mathbf{S} \mathbf{V}^{\top} = \mathbf{U} \mathbf{S} \mathbf{V}^{\top} = \mathbf{X}$$

means that $\mathbf{G}_0 \in \mathcal{S}_{\mathrm{unsup}}^{\top}(k)$ is one interpolating solution, since $(\mathbf{I}_d - \mathbf{G}_0 \mathbf{G}_0^{\top})\mathbf{X} = 0$. We will use $\mathbf{G}_0$ to generate more solutions. In fact, for each real orthogonal matrix $\mathbf{U} \in \mathbb{R}^{k \times k}$, we see that $\mathbf{G}_0 \mathbf{U} \in \mathcal{S}_{\mathrm{unsup}}^{\top}(k)$ because $(\mathbf{G}_0 \mathbf{U})(\mathbf{G}_0 \mathbf{U})^{\top} = \mathbf{G}_0 \mathbf{U} \mathbf{U}^{\top} \mathbf{G}_0^{\top} = \mathbf{G}_0 \mathbf{G}_0^{\top}$. We can identify solutions of the form $\mathbf{G}_0 \mathbf{U}$ with the orthogonal group $O(k)$ which is a smooth manifold (in particular, a real Lie group) of dimension $\frac{k(k-1)}{2}$ (Lee, 2003). We define $\mathcal{S}_{\mathrm{orth}}(\mathbf{G}_0) = \{\mathbf{G} \in \mathbb{R}^{d \times k} : \mathbf{G} = \mathbf{G}_0 \mathbf{U}, \mathbf{U} \in O(k)\}$ and maintain that $\mathcal{S}_{\mathrm{orth}}(\mathbf{G}_0) \subset \mathcal{S}_{\mathrm{unsup}}^{\top}(k)$.

Now, we consider interpolating solutions for when $n = k$ and show that $\mathcal{S}_{\mathrm{orth}}(\mathbf{G}_0) \supset \mathcal{S}_{\mathrm{unsup}}^{\top}(k)$ so that $\mathcal{S}_{\mathrm{unsup}}^{\top}(k)$ is a smooth manifold of dimension $\frac{k(k-1)}{2}$. Let $\mathbf{G} \in \mathcal{S}_{\mathrm{unsup}}^{\top}(k)$ be given. For any $\mathbf{x} \in \mathrm{span}\{\mathbf{x}_1, \ldots, \mathbf{x}_n\}$, it is clear that $\mathbf{G}\mathbf{G}^{\top}\mathbf{x} = \mathbf{x}$. Moreover, since $\mathbf{G}\mathbf{G}^{\top}$ has rank $n = k$, we see that any $\mathbf{y}$ that is independent of our samples must be in the null space of $\mathbf{G}\mathbf{G}^{\top}$ so that $\mathbf{G}\mathbf{G}^{\top}\mathbf{y} = 0$. Thus, $\mathbf{G}\mathbf{G}^{\top}$ is an orthogonal projection matrix (Axler, 1997). Thus, $\mathrm{range}(\mathbf{G}\mathbf{G}^{\top}) = \mathrm{span}(\mathbf{x}_1, \ldots, \mathbf{x}_n)$ and the eigenvalues of $\mathbf{G}\mathbf{G}^{\top}$ are all either 0 or 1.

Let $\mathbf{G} = \mathbf{U}\mathbf{S}\mathbf{V}^{\top}$ be $\mathbf{G}_0 = \mathbf{U}_0 \mathbf{S}\mathbf{V}_0^{\top}$ two singular value decompositions; note that the singular value matrix $\mathbf{S} = \begin{bmatrix} \mathbf{I}_k \\ \mathbf{0}_{d-k \times k} \end{bmatrix}$ is the same in these two because they have the same singular values. Then, we define

$\mathbf{W} = \mathbf{G}_0^\top \mathbf{G}$ so that

$$\begin{aligned}
\mathbf{G}_0 \mathbf{W} &= \mathbf{G}_0 \mathbf{G}_0^\top \mathbf{G} \\
&= \mathbf{G} \mathbf{G}^\top \mathbf{G} \\
&= \mathbf{G} \mathbf{V} \mathbf{S}^\top \mathbf{U}^\top \mathbf{U} \mathbf{S} \mathbf{V}^\top \\
&= \mathbf{G}. && \text{(Since } \mathbf{S}^\top \mathbf{S} = \mathbf{I}_n)
\end{aligned}$$

Note that $\mathbf{W}\mathbf{W}^\top = \mathbf{G}_0^\top \mathbf{G} \mathbf{G}^\top \mathbf{G}_0 = \mathbf{G}_0^\top \mathbf{G}_0 \mathbf{G}_0^\top \mathbf{G}_0 = \mathbf{I}_k$ and $\mathbf{W}^\top \mathbf{W} = \mathbf{G}^\top \mathbf{G}_0 \mathbf{G}_0^\top \mathbf{G} = \mathbf{G}^\top \mathbf{G} \mathbf{G}^\top \mathbf{G} = \mathbf{I}_k$ imply that $\mathbf{W} \in O(k)$ is a real orthogonal matrix. Thus, any arbitrary interpolating solution $\mathbf{G}$ can be written as $\mathbf{G} = \mathbf{G}_0 \mathbf{U}$ for $\mathbf{U} \in O(k)$. This means that $\mathcal{S}_{\text{orth}}(\mathbf{G}_0) \supset \mathcal{S}_{\text{unsup}}^\top(k)$ implying that $\mathcal{S}_{\text{orth}}(\mathbf{G}_0) = \mathcal{S}_{\text{unsup}}^\top(k)$, as desired. Thus, $\mathcal{S}_{\text{unsup}}^\top(k)$ is a smooth manifold with dimension $\frac{k(k-1)}{2} = \frac{n(n-1)}{2}$ when $n = k$.

Now we consider the $k > n$ case. For any interpolating $\mathbf{G} \in \mathcal{S}_{\text{unsup}}^\top(k)$, we must have that $n$ columns of $\mathbf{G}$ span the column space of $\mathbf{X}$. That leaves $k - n$ columns of $\mathbf{G}$, each in $\mathbb{R}^d$, free. Suppose that the first $n$ columns of $\mathbf{G}$ are the ones which span the column space of $\mathbf{X}$ and the next $k - n$ columns are arbitrary; we will write $\mathbf{G} = \begin{bmatrix} \mathbf{G}_{\mathbf{X}} & \mathbf{G}_a \end{bmatrix}$. Thus, we have that $\mathbf{G}\mathbf{G}^\top = \mathbf{G}_{\mathbf{X}} \mathbf{G}_{\mathbf{X}}^\top + \mathbf{G}_a \mathbf{G}_a^\top$, meaning that we only interpolate if $\text{range}(\mathbf{G}_a) \cap \text{span}(\mathbf{x}_1, \ldots, \mathbf{x}_n) = \{0\}$. Of course, by the arguments made above, we see that $\mathbf{G}_{\mathbf{X}} \mathbf{U}$ will work instead of $\mathbf{G}_{\mathbf{X}}$ for any $\mathbf{U} \in O(n)$, this means that for an appropriate selection of $\mathbf{G}_a$, we have that $\{\mathbf{G} \in \mathbb{R}^{d \times k} : L_{\text{unsup}}^\top(\mathbf{G}, \mathbf{X}) = 0, \mathbf{G} = \begin{bmatrix} \mathbf{G}_{\mathbf{X}} & \mathbf{G}_a \end{bmatrix}, \mathbf{G}_a \text{ fixed}\}$ has dimension $\frac{n(n-1)}{2}$. Since $\mathbf{G}_a$ can be arbitrary (outside of the column span of $\mathbf{X}$), it is of dimension $(k-n)(d-n)$. To see this, take $k = n+1$. Then, $\mathbf{G}_a$ is a single vector which can span a $d - n$ dimensional subspace of $\mathbb{R}^d$. If $k = n+2$, both vectors can span the $d - n$-dimensional subspace of $\mathbb{R}^d$ making them $2(d-n) = (k-n)(d-n)$-dimensional. Thus, we see that $\{\mathbf{G} \in \mathbb{R}^{d \times k} : L_{\text{unsup}}^\top(\mathbf{G}, \mathbf{X}) = 0, \mathbf{G} = \begin{bmatrix} \mathbf{G}_{\mathbf{X}} & \mathbf{G}_a \end{bmatrix}\}$ must have dimension $\frac{n(n-1)}{2}(k-n)(d-n)$. This doesn't completely characterize $\mathcal{S}_{\text{unsup}}^\top(k)$ because we fixed the structure of $\mathbf{G}$.

Suppose that $\mathbf{G} \in \mathcal{S}_{\text{unsup}}^\top(k)$ is an interpolating solution. Then, any $n$ columns of $\mathbf{G}$ can span the column space of $\mathbf{X}$ and the remaining $k - n$ columns are free. For each such combination of data-spanning and free columns, we get a smooth manifold of dimension $\frac{n(n-1)}{2}(k-n)(d-n)$. Thus, $\mathcal{S}_{\text{unsup}}^\top(k)$ is the union of $\binom{n}{k}$ smooth manifolds of dimension $\frac{n(n-1)}{2}(k-n)(d-n)$. One can check that each of these manifolds is disjoint, and hence this union results in another smooth manifold of the same dimension. For if this is not true, the same $\mathbf{G}$ is in two combinations of data-spanning and free column configurations. Since the data spanning vectors are non-zero and linearly independent of the free vectors, this is a contradiction. In summary, $\mathcal{S}_{\text{unsup}}^\top(k)$ is a smooth manifold of dimension $\frac{n(n-1)}{2}(k-n)(d-n)$.

$\square$

**Theorem 3.** *Suppose that $\mathbf{X} \in \mathbb{R}^{d \times n}$ has full rank of $\min\{d, n\}$. For the unsupervised loss $L_{unsup}^\dagger(\mathbf{G}, \mathbf{X}) \triangleq \|(\mathbf{I}_d - \mathbf{G}\mathbf{G}^\dagger)\mathbf{X}\|_F^2$, let $\mathcal{S}_{unsup}^\dagger(k) \triangleq \{\mathbf{G} \in \mathbb{R}^{d \times k} : L_{unsup}^\dagger(\mathbf{G}, \mathbf{X}) = 0\}$ be the set of interpolating solutions. Then,*

1. *$\mathcal{S}_{unsup}^\dagger(k) = \emptyset$ if $n > k$.*

2. *$\mathcal{S}_{unsup}^\dagger(k)$ is a smooth manifold of dimension $n^2$ when $n = k$.*

3. *$\mathcal{S}_{unsup}^\dagger(k)$ is the union of $\binom{n}{k}$ smooth manifolds of dimension $n^2(k-n)d$ when $k > n$.*

*Proof for Theorem 3.* Suppose that $n > k$ and, for the sake of a contradiction, that $\mathcal{S}_{\text{unsup}}^\dagger(k) \neq \emptyset$. This means that there exists a $\mathbf{G} \in \mathcal{S}_{\text{unsup}}^\dagger(k)$ so that $\|(\mathbf{I}_d - \mathbf{G}\mathbf{G}^\dagger)\mathbf{X}\|_F^2 = 0$. Thus, for each column $\mathbf{x}_i$ of $\mathbf{X}$ we have that $\|(\mathbf{I}_d - \mathbf{G}\mathbf{G}^\dagger)\mathbf{x}_i\|_2^2 = 0$. In other words, we know that $\mathbf{G}\mathbf{G}^\dagger \mathbf{x}_i = \mathbf{x}_i$ implies that $\mathbf{G}\mathbf{G}^\dagger$ has at least $n$ eigenvalues of 1 corresponding to $n$ eigenvectors since the samples are linearly independent. By the spectral theorem (Theorem 2.5.6 in Horn & Johnson (2012)), $\mathbf{G}\mathbf{G}^\dagger$ is diagonalizable and thus $\text{rank}(\mathbf{G}\mathbf{G}^\dagger) \geq n$. However, we know that $\text{rank}(\mathbf{G}\mathbf{G}^\dagger) \leq k$ by simple rank inequalities (0.4.5 (a) in Horn & Johnson (2012)), a contradiction. Therefore, we know that $\mathcal{S}_{\text{unsup}}^\dagger(k) = \emptyset$.

Now suppose that $n = k$. Let $\mathbf{G}_0 = \mathbf{X}$. Clearly, $\mathbf{G}_0$ has full column rank and thus has an explicit pseudo-inverse formulation. Then, we see that

$$\mathbf{G}_0 \mathbf{G}_0^\dagger \mathbf{X} = \mathbf{G}_0 (\mathbf{G}_0^\top \mathbf{G}_0)^{-1} \mathbf{G}_0^\top \mathbf{X} = \mathbf{X} (\mathbf{X}^\top \mathbf{X})^{-1} \mathbf{X}^\top \mathbf{X} = \mathbf{X}$$

means that $\mathbf{G}_0 \in \mathcal{S}_{\text{unsup}}^\dagger(k)$ is one interpolating solution, since $(\mathbf{I}_d - \mathbf{G}_0 \mathbf{G}_0^\dagger)\mathbf{X} = 0$. We will use $\mathbf{G}_0$ to generate more solutions. In fact, for each invertible matrix $\mathbf{A} \in \mathbb{R}^{k \times k}$, we see that $\mathbf{G}_0 \mathbf{A} \in \mathcal{S}_{\text{unsup}}^\dagger(k)$ because $(\mathbf{G}_0 \mathbf{A})(\mathbf{G}_0 \mathbf{A})^\dagger = \mathbf{G}_0 \mathbf{G}_0^\dagger$. We can identify solutions of the form $\mathbf{G}_0 \mathbf{A}$ with the real general linear group $\text{GL}(k)$ which is a smooth manifold (in particular, a real Lie group) of dimension $n^2$ Lee (2003). We define $\mathcal{S}_{\text{GL}}(\mathbf{G}_0) = \{\mathbf{G} \in \mathbb{R}^{d \times k} : \mathbf{G} = \mathbf{G}_0 \mathbf{A}, \mathbf{A} \in \text{GL}(k)\}$ and maintain that $\mathcal{S}_{\text{GL}}(\mathbf{G}_0) \subset \mathcal{S}_{\text{unsup}}^\dagger(k)$.

Now, we consider interpolating solutions for when $n = k$ and show that $\mathcal{S}_{\text{GL}}(\mathbf{G}_0) \supset \mathcal{S}_{\text{unsup}}^\dagger(k)$ so that $\mathcal{S}_{\text{unsup}}^\dagger(k)$ is a smooth manifold of dimension $n^2$. Let $\mathbf{G} \in \mathcal{S}_{\text{unsup}}^\dagger(k)$ be given. For any $\mathbf{x} \in \text{span}\{\mathbf{x}_1, \ldots, \mathbf{x}_n\}$, it is clear that $\mathbf{G}\mathbf{G}^\dagger \mathbf{x} = \mathbf{x}$. Moreover, since $\mathbf{G}\mathbf{G}^\dagger$ has rank $n = k$, we see that any $\mathbf{y}$ that is independent of our samples must be in the null space of $\mathbf{G}\mathbf{G}^\dagger$ so that $\mathbf{G}\mathbf{G}^\dagger \mathbf{y} = 0$. Thus, $\mathbf{G}\mathbf{G}^\dagger$ is an orthogonal projection matrix Axler (1997). Thus, $\text{range}(\mathbf{G}\mathbf{G}^\dagger) = \text{span}(\mathbf{x}_1, \ldots, \mathbf{x}_n)$ and the eigenvalues of $\mathbf{G}\mathbf{G}^\dagger$ are all either 0 or 1. Note that since $\mathbf{G}\mathbf{G}^\dagger$ and $\mathbf{G}_0 \mathbf{G}_0^\dagger$ have rank $n$ and are projection matrices onto the the column space of $\mathbf{X}$, we have that $\mathbf{G}\mathbf{G}^\dagger = \mathbf{G}_0 \mathbf{G}_0^\dagger$. Now we define $\mathbf{A} = \mathbf{G}_0^\dagger \mathbf{G}$ so that

$$\begin{aligned} \mathbf{G}_0 \mathbf{A} &= \mathbf{G}_0 \mathbf{G}_0^\dagger \mathbf{G} \\ &= \mathbf{G}\mathbf{G}^\dagger \mathbf{G} \\ &= \mathbf{G} \end{aligned}$$

Note that $\mathbf{A} \in \text{GL}(k)$ is invertible, otherwise $\mathbf{G}$ would be rank deficient and not able to span the column space of $\mathbf{X}$. Thus, any arbitrary interpolating solution $\mathbf{G}$ can be written as $\mathbf{G} = \mathbf{G}_0 \mathbf{A}$ for $\mathbf{A} \in \text{GL}(k)$. This means that $\mathcal{S}_{\text{GL}}(\mathbf{G}_0) \supset \mathcal{S}_{\text{unsup}}^\dagger(k)$ implying that $\mathcal{S}_{\text{GL}}(\mathbf{G}_0) = \mathcal{S}_{\text{unsup}}^\dagger(k)$, as desired. Thus, $\mathcal{S}_{\text{unsup}}^\dagger(k)$ is a smooth manifold with dimension $n^2$ when $n = k$.

Now we consider the $k > n$ case. For any interpolating $\mathbf{G} \in \mathcal{S}_{\text{unsup}}^\dagger(k)$, we must have that $n$ columns of $\mathbf{G}$ span the column space of $\mathbf{X}$. That leaves $k - n$ columns of $\mathbf{G}$, each in $\mathbb{R}^d$, free to be arbitrary. Suppose that the first $n$ columns of $\mathbf{G}$ are the ones which span the column space of $\mathbf{X}$ and the next $k - n$ columns are arbitrary; we will write $\mathbf{G} = \begin{bmatrix} \mathbf{G_X} & \mathbf{G}_a \end{bmatrix}$. In this case, $\mathbf{G}_a$ can be completely arbitrary and even contain columns which span the data. To see this, let $\mathbf{G}_a$ be arbitrary and note that $\mathbf{G}\mathbf{G}^\dagger \mathbf{G} = \mathbf{G}$ for any pseudo-inverse (even if we don't have full rank). Thus,

$$\mathbf{G}\mathbf{G}^\dagger \mathbf{G} = \mathbf{G}\mathbf{G}^\dagger \begin{bmatrix} \mathbf{G_X} & \mathbf{G}_a \end{bmatrix} = \begin{bmatrix} \mathbf{G}\mathbf{G}^\dagger \mathbf{G_X} & \mathbf{G}\mathbf{G}^\dagger \mathbf{G}_a \end{bmatrix} = \mathbf{G} = \begin{bmatrix} \mathbf{G_X} & \mathbf{G}_a \end{bmatrix}$$

implies that $\mathbf{G}\mathbf{G}^\dagger \mathbf{G_X} = \mathbf{G_X}$. Since $\mathbf{G_X}$ has the form $\mathbf{G_X} = \mathbf{XA}$ for $\mathbf{A} \in \text{GL}(k)$, we see that $\mathbf{G}\mathbf{G}^\dagger \mathbf{XA} = \mathbf{XA}$. Finally, since $\mathbf{A}$ is invertible, we have that $\mathbf{G}\mathbf{G}^\dagger \mathbf{X} = \mathbf{X}$.

Since $\mathbf{G_X A}$ will work instead of $\mathbf{G_X}$ for any $\mathbf{A} \in \text{GL}(n)$, this means that for fixed $\mathbf{G}_a$, we have that $\{\mathbf{G} \in \mathbb{R}^{d \times k} : \|(\mathbf{I}_d - \mathbf{G}\mathbf{G}^\dagger)\mathbf{X}\| = 0, \mathbf{G} = \begin{bmatrix} \mathbf{G_X} & \mathbf{G}_a \end{bmatrix}, \mathbf{G}_a \text{ fixed}\}$ has dimension $n^2$. Since $\mathbf{G}_a$ can be arbitrary and is of dimension $(k - n)d$, we see that $\{\mathbf{G} \in \mathbb{R}^{d \times k} : \|(\mathbf{I}_d - \mathbf{G}\mathbf{G}^\dagger)\mathbf{X}\| = 0, \mathbf{G} = \begin{bmatrix} \mathbf{G_X} & \mathbf{G}_a \end{bmatrix}\}$ must have dimension $n^2(k - n)d$. This doesn't completely characterize $\mathcal{S}_{\text{unsup}}^\dagger(k)$ because we fixed the structure of $\mathbf{G}$.

Suppose that $\mathbf{G} \in \mathcal{S}_{\text{unsup}}^\dagger(k)$ is an interpolating solution. Then, any $n$ columns of $\mathbf{G}$ can span the column space of $\mathbf{X}$ and the remaining $k - n$ columns can be arbitrary. For each such combination of data-spanning and arbitrary columns, we get a smooth manifold of dimension $n^2(k - n)d$. Thus, $\mathcal{S}_{\text{unsup}}^\dagger(k)$ is the union of $\binom{n}{k}$ smooth manifolds of dimension $n^2(k - n)d$. However, $\mathcal{S}_{\text{unsup}}^\dagger(k)$ need not be a manifold because for one configuration the arbitrary columns of $\mathbf{G}$ can equal the data-spanning columns of $\mathbf{G}$ for another configuration. This results in self-intersection. For a concrete example, let $n = 1, k = 2$. Then, for the two configurations $\begin{bmatrix} \mathbf{G_x} & \mathbf{G}_a \end{bmatrix}\}$ and $\begin{bmatrix} \mathbf{G}_a & \mathbf{G_x} \end{bmatrix}\}$, we can have the solution $\begin{bmatrix} \mathbf{G_x} & \mathbf{G_x} \end{bmatrix}\}$. In summary, $\mathcal{S}_{\text{unsup}}^\dagger(k)$ is the union of $\binom{n}{k}$ smooth manifolds of dimension $n^2(k - n)d$. $\qquad\square$

## C.2 The pseudo-supervised solutions

**Theorem 2.** *Suppose that $\mathbf{X} \in \mathbb{R}^{d \times n}$ has full rank of $\min\{d, n\}$ and let $\lambda > 0$ be given. For the pseudo-supervised loss $L_{ps}^{\top}(\mathbf{G}, \mathbf{X}; \lambda) \triangleq \frac{\lambda}{n_{ps}} \|\mathbf{G}\mathbf{Z}_{\mathcal{S}}^{ps} - \mathbf{X}^{ps}\|_F^2 + \frac{1}{n}\|(\mathbf{I}_d - \mathbf{G}\mathbf{G}^{\top})\mathbf{X}\|_F^2$, let $\mathcal{S}_{ps}^{\top}(k) \triangleq \{\mathbf{G} \in \mathbb{R}^{d \times k} : L_{ps}^{\top}(\mathbf{G}, \mathbf{X}, \lambda) = 0\}$ be the set of interpolating solutions. Then,*

1. *$\mathcal{S}_{ps}^{\top}(k) = \emptyset$ if $n > k$ and $\mathbf{Z} \in \mathbb{R}^{k \times n}$ is arbitrary.*

2. *$\mathcal{S}_{ps}^{\top}(k)$ has only one element if $n = k$ and $\mathbf{Z} \in \mathbb{R}^{k \times n}$ is given so that $\mathbf{Z}^{\top}\mathbf{Z} = \mathbf{X}^{\top}\mathbf{X}$.*

3. *$\mathcal{S}_{ps}^{\top}(k)$ is the union of $\binom{n}{k}$ smooth manifolds of dimension $(k-n)(d-n)$ if $k > n$ and $\mathbf{Z} = \begin{bmatrix} \mathbf{Z}_1 \\ \mathbf{0} \end{bmatrix} \in \mathbb{R}^{k \times n}$ is given so that $\mathbf{Z}_1^{\top}\mathbf{Z}_1 = \mathbf{X}^{\top}\mathbf{X}$.*

*Proof for Theorem 2.* Since $\mathcal{S}_{\mathrm{ps}}^{\top}(k) \subset \mathcal{S}_{\mathrm{unsup}}^{\top}(k)$, this implies that $\mathcal{S}_{\mathrm{ps}}^{\top}(k) = \emptyset$ when $n > k$.

Suppose that $n = k$ and from Theorem 1 we see that the solutions to $L_{\mathrm{ps}}^{\top}(\mathbf{G}, \mathbf{X}, \lambda)$ must have the form of $\mathbf{U}_k \mathbf{W}$ for $\mathbf{W} \in O(k)$. From the condition $\mathbf{Z}^{\top}\mathbf{Z} = \mathbf{X}^{\top}\mathbf{X}$, we see that $\mathbf{Z} = \mathbf{A}\mathbf{D}\mathbf{V}^{\top}$ with a unitary matrix $\mathbf{A}$ and the other matrices from the SVD of $\mathbf{X} = \mathbf{U} \begin{bmatrix} \mathbf{D} \\ \mathbf{0} \end{bmatrix} \mathbf{V}^{\top}$; this joint decomposition can be derived from the fact that our condition implies that the singular values and right singular vectors of $\mathbf{Z}$ and $\mathbf{X}$ are the same. Thus, we see that

$$
\begin{aligned}
\mathbf{G}\mathbf{Z} - \mathbf{X} &= \mathbf{U}_k \mathbf{W} \mathbf{Z} - \mathbf{U} \begin{bmatrix} \mathbf{D} \\ \mathbf{0} \end{bmatrix} \mathbf{V}^{\top} \\
&= \mathbf{U}_k \mathbf{W} \mathbf{A} \mathbf{D} \mathbf{V}^{\top} - \mathbf{U} \begin{bmatrix} \mathbf{D} \\ \mathbf{0} \end{bmatrix} \mathbf{V}^{\top} && (\mathbf{Z}^{\top}\mathbf{Z} = \mathbf{X}^{\top}\mathbf{X}) \\
&= \mathbf{U}_k \mathbf{W} \mathbf{A} \mathbf{D} \mathbf{V}^{\top} - \mathbf{U}_k \mathbf{D} \mathbf{V}^{\top} \\
&= \mathbf{U}_k \left( \mathbf{W}\mathbf{A} - \mathbf{I}_k \right) \mathbf{D} \mathbf{V}^{\top} \\
&= \mathbf{0}
\end{aligned}
$$

if and only if $\mathbf{W} = \mathbf{A}$, which is permissible since both are unitary. Hence, we have only one solution.

Now suppose that $k > n$. Then, we have that our solution from Theorem 1 has the form of $\mathbf{G} = \begin{bmatrix} \mathbf{U}_n \mathbf{W} & \mathbf{G}_a \end{bmatrix}$, where $\mathbf{W} \in O(n)$ and $\mathbf{G}_a$ is arbitrary. Then,

$$
\begin{aligned}
\mathbf{G}\mathbf{Z} - \mathbf{X} &= \begin{bmatrix} \mathbf{U}_n \mathbf{W} & \mathbf{G}_a \end{bmatrix} \begin{bmatrix} \mathbf{A}\mathbf{D}\mathbf{V}^{\top} \\ \mathbf{0} \end{bmatrix} - \mathbf{U}_n \mathbf{D} \mathbf{V}^{\top} \\
&= \mathbf{U}_n \mathbf{W} \mathbf{A} \mathbf{D} \mathbf{V}^{\top} - \mathbf{U}_n \mathbf{D} \mathbf{V}^{\top} \\
&= 0
\end{aligned}
$$

if and only if $\mathbf{W} = \mathbf{A}^{\top}$. So we have one solution in this case, but since $\mathbf{G}_a$ is arbitrary, we have $(k-n)(d-n)$ solutions. Moreover, since the columns of $\mathbf{G}$ were chosen in this convenient way, we actually have a union of $\binom{n}{k}$ solutions spaces of dimension $(k-n)(d-n)$. $\qquad \square$

**Theorem 4.** *Suppose that $\mathbf{X} \in \mathbb{R}^{d \times n}$ has full rank of $\min\{d, n\}$ and let $\lambda > 0$ be given. $L_{ps}^{\dagger}(\mathbf{G}, \mathbf{X}; \lambda) \triangleq \frac{\lambda}{n_{ps}} \|\mathbf{G}\mathbf{Z}_{\mathcal{S}}^{ps} - \mathbf{X}^{ps}\|_F^2 + \frac{1}{n}\|(\mathbf{I}_d - \mathbf{G}\mathbf{G}^{\dagger})\mathbf{X}\|_F^2$, let $\mathcal{S}_{ps}^{\dagger}(k) \triangleq \{\mathbf{G} \in \mathbb{R}^{d \times k} : L_{ps}^{\dagger}(\mathbf{G}, \mathbf{X}, \lambda) = 0\}$ be the set of interpolating solutions. Then,*

1. *$\mathcal{S}_{ps}^{\dagger}(k) = \emptyset$ if $n > k$.*

2. *$\mathcal{S}_{ps}^{\dagger}(k)$ has only one element when $n = k$.*

| | $\mathcal{S}_{\text{unsup}}^{\top}$ | $\mathcal{S}_{\text{unsup}}^{\dagger}$ | $\mathcal{S}_{\text{ps}}^{\dagger}$ |
|---|---|---|---|
| $n < k$ | $0$ | $0$ | $0$ |
| $n = k$ | $\frac{n(n-1)}{2}$ | $n^2$ | $1$ |
| $k > n$ | $\frac{n(n-1)}{2}(k-n)(d-n)$ | $n^2(k-n)d$ | $(k-n)d$ |

**Table 1:** The size of the solution sets

3. $\mathcal{S}_{ps}^{\dagger}(k)$ *an affine space of dimension* $(k-n)d$ *when* $k > n$.

*Proof for Theorem 4.* Clearly, $\mathcal{S}_{\text{ps}}^{\dagger} \subset \mathcal{S}_{\text{unsup}}^{\top}$, implying that $\mathcal{S}_{\text{ps}}^{\dagger} = \emptyset$ when $n > k$.

When $n = k$, we see that the pseudo-supervised term reaches zero only when we have the unique solution of $\mathbf{G} = \mathbf{X}\mathbf{Z}^{-1}$. Note that for this selction of $\mathbf{G}$ we have that

$$\mathbf{G}\mathbf{G}^{\dagger}\mathbf{X} = \mathbf{X}\mathbf{Z}^{-1}(\mathbf{Z}^{-\top}\mathbf{X}^{\top}\mathbf{X}\mathbf{Z}^{-1})^{-1}\mathbf{Z}^{-\top}\mathbf{X}^{\top}\mathbf{X} = \mathbf{X}(\mathbf{X}^{\top}\mathbf{X})^{-1}\mathbf{X}^{\top}\mathbf{X} = \mathbf{X},$$

meaning that $\mathbf{G} = \mathbf{X}\mathbf{Z}^{-1}$ is the only unique solution to the problem when $n = k$.

Now suppose that $k > n$ and let $\mathbf{Z} = \mathbf{U}\mathbf{S}\mathbf{V}^{\top}$ be decomposed via SVD, where $\mathbf{S} = \begin{bmatrix} \boldsymbol{\Sigma} \\ \mathbf{0}_{k-n \times n} \end{bmatrix}$ with invertible $\boldsymbol{\Sigma} \in \mathbb{R}^{n \times n}$. For an arbitrary solution $\mathbf{G}$ we can write it as $\mathbf{G} = \begin{bmatrix} \mathbf{G}' & \mathbf{G}_a \end{bmatrix} \mathbf{U}^{\top}$ for $\mathbf{G}' \in \mathbb{R}^{d \times n}, \mathbf{G}_a \in \mathbb{R}^{d \times k-n}$. Note that since we are multiplying by $\mathbf{U}^{\top}$, which is invertible, this does not change the solution set we are interested in but merely rotates it. Hence, we have that

$$\mathbf{X} = \mathbf{G}\mathbf{Z} = \begin{bmatrix} \mathbf{G}' & \mathbf{G}_a \end{bmatrix} \mathbf{U}^{\top}\mathbf{U}\mathbf{S}\mathbf{V}^{\top} = \begin{bmatrix} \mathbf{G}' & \mathbf{G}_a \end{bmatrix} \begin{bmatrix} \boldsymbol{\Sigma} \\ \mathbf{0} \end{bmatrix} \mathbf{V}^{\top} = \mathbf{G}'\boldsymbol{\Sigma}\mathbf{V}^{\top}.$$

Thus, $\mathbf{G}' = \mathbf{X}\mathbf{V}\boldsymbol{\Sigma}^{-1}$ uniquely since both $\boldsymbol{\Sigma}$ and $\mathbf{V}^{\top}$ are invertible. On the other hand, $\mathbf{G}_a$ can be anything. Thus, $\{\mathbf{G} \in \mathbb{R}^{d \times k} : \|\mathbf{G}\mathbf{Z} - \mathbf{X}\|_F^2 = 0\}$ can be identified with the matrices $\mathbf{G}_a \in \mathbb{R}^{d \times k-n}$. Moreover, note that any solution of the form $\mathbf{G} = \begin{bmatrix} \mathbf{X}\mathbf{V}\boldsymbol{\Sigma}^{-1} & \mathbf{G}_a \end{bmatrix} \mathbf{U}^{\top}$ must have that $\mathbf{G}\mathbf{G}^{\dagger}\mathbf{X}\mathbf{V}\boldsymbol{\Sigma}^{-1} = \mathbf{X}\mathbf{V}\boldsymbol{\Sigma}^{-1}$ as shown in the proof of Theorem 3. Thus, $\mathbf{G}\mathbf{G}^{\dagger}\mathbf{X} = \mathbf{X}$ meaning that $\mathbf{G} \in \mathcal{S}_{\text{ps}}^{\dagger}(k)$. Thus, $\mathcal{S}_{\text{ps}}^{\dagger}(k)$ is an affine space of dimension $d(k-n)$.

$\square$

Recall that in the proof of Theorem 3 we have that the part of $\mathbf{G}$ which contributes has the form $\mathbf{X}\mathbf{A}$ for any invertible $\mathbf{A}$. We see here that the pseudo-supervision forces $\mathbf{A}$ to be equal to $\mathbf{V}\boldsymbol{\Sigma}^{-1}$ (which are completely constructed from $\mathbf{Z}$) and removes all the degrees of freedom in $\mathbf{A}$. We summarize these results in the Table 1.

# D   Experiments on linear models and gradient details

## D.1   Details regarding linear experiments

In the linear setting, we set $\boldsymbol{\Gamma} \in \mathbb{R}^{d \times m}$ to be the first $m = 10$ columns of a Hadamard matrix multiplied by $\frac{1}{\sqrt{d}}$, where $d = 64$. Trails using random orthonormal columns for $\boldsymbol{\Gamma}$ yielded extremely similar results, therefore we only show plots for the Hadamard $\boldsymbol{\Gamma}$. Then, we create our data by drawing $n = 20$ samples from $\boldsymbol{\Gamma}\mathbf{z} + \boldsymbol{\epsilon}$, where $\mathbf{z} \sim \mathcal{N}(\mathbf{0}, \mathbf{I}_m)$ and $\boldsymbol{\epsilon} = \mathcal{N}(\mathbf{0}, 0.15^2\mathbf{I}_d)$. Our initial matrix $\mathbf{G} \in \mathbb{R}^{d \times k}$ is drawn from an isotropic Gaussian with 0.03 standard deviation. We have $k \in \{1, 3, 5, \ldots, 127\}$ for the pseudo-supervised experiments and $k \in \{1, 2, \ldots, 40\}$ for the supervised experiments. For all these experiments, we have $n_{\text{ps}}$ and $n_{\text{sup}}$ take values in $\{0, 2, 4, 12, 18, 20\}$.

We perform gradient descent with a maximum of 500 iterations. The initial step size is 0.0001 after which we adaptively pick the current iteration's step size which will reduce the training loss most. We do this by multiplying the current step size by values in $\{0.0000001, 0.000005, 0.000001, 0.00001, 0.0001, 0.001, 0.01, 0.1, 1, 10, 100\}$ and picking the value which will

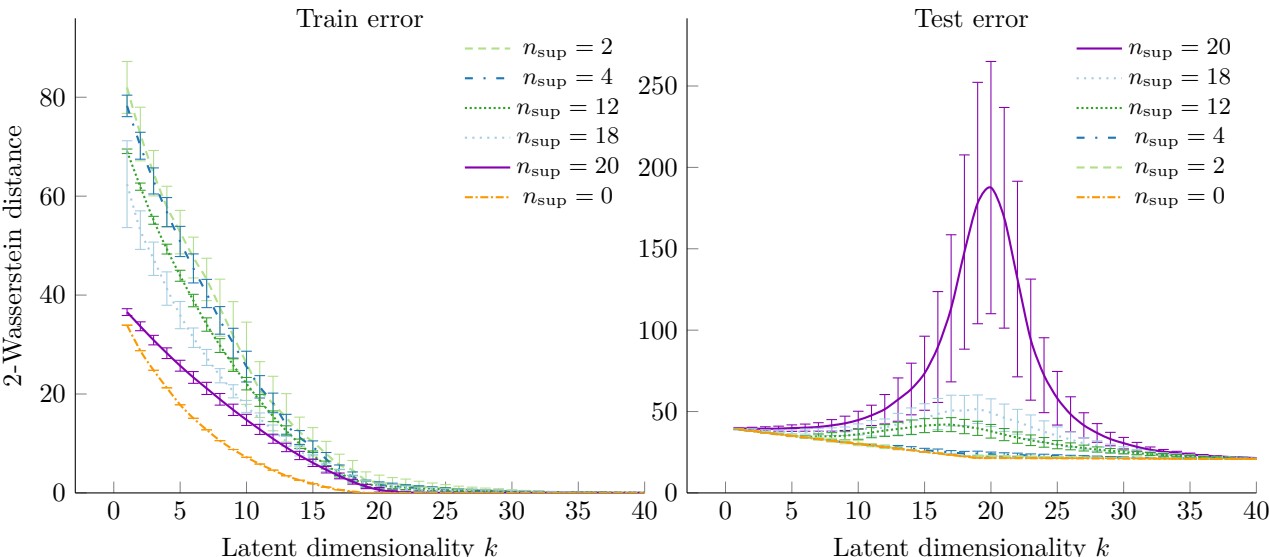

**Figure 8:** In this figure, we minimize the loss in Equation (3). The legends are displayed in the same order as the curves appear on the plot for clarity. This figure is a more detailed version of Figure 2. These plots are averaged over 200 experiments, so we plot the standard deviation instead of the standard error (which would be about 14 times smaller).

yield the lowest training loss. If the matrix $\mathbf{G}$ does not change more than 0.00001 in Frobenius norm for more than 5 iterations, then the optimization also stops. If the Frobenius norm of the gradient is less than 0.05, then the optimization stops. The gradients are calculated in Appendix D.2.

We run all these experiments 200 times and average the results. For each experiment, we pick a new seed and re-run the same script. Therefore, the pseudo-supervised examples are fixed for each experiment as we vary $k$ and $n_{\mathrm{ps}}$. Hence, the errorbars in Figures 8 to 12 show one standard deviation of how the choice of matrix initialization, pseudo-supervision samples, and data samples all affect the test error.

## D.2   Gradient calculations

The losses introduced in Equations (4) to (6) all have similar forms, so we only show what is the gradient for Equation (4) and the other ones are easily obtained. For completeness, we restate the loss:

$$\mathcal{L}^{\mathrm{train}}(\mathbf{G}, \mathcal{D}) = \frac{1}{n_{\mathrm{ps}}}\|\mathbf{G}\mathbf{Z}_{\mathcal{S}}^{\mathrm{ps}} - \mathbf{X}^{\mathrm{ps}}\|_F^2 + \frac{1}{n_{\mathrm{unsup}}}\|(\mathbf{I}_d - \mathbf{G}\mathbf{G}^\top)\mathbf{X}^{\mathrm{unsup}}\|_F^2.$$

The gradient of the first term is

$$\nabla_{\mathbf{G}}\frac{1}{n_{\mathrm{ps}}}\|\mathbf{G}\mathbf{Z}_{\mathcal{S}}^{\mathrm{ps}} - \mathbf{X}^{\mathrm{ps}}\|_F^2 = \frac{1}{n_{\mathrm{ps}}}\nabla_{\mathbf{G}}\|(\mathbf{Z}_{\mathcal{S}}^{\mathrm{ps}})^\top\mathbf{G}^\top - (\mathbf{X}^{\mathrm{ps}})^\top\|_F^2 \qquad \text{(Frobenius transpose invariance)}$$

$$= \frac{1}{n_{\mathrm{ps}}}\left(\nabla_{\mathbf{G}^\top}\|(\mathbf{Z}_{\mathcal{S}}^{\mathrm{ps}})^\top\mathbf{G}^\top - (\mathbf{X}^{\mathrm{ps}})^\top\|_F^2\right)^\top \qquad \text{(Section 4.2.3 of (Gentle, 2007))}$$

$$= \frac{1}{n_{\mathrm{ps}}}\left(2\mathbf{Z}_{\mathcal{S}}^{\mathrm{ps}}\big((\mathbf{Z}_{\mathcal{S}}^{\mathrm{ps}})^\top\mathbf{G}^\top - (\mathbf{X}^{\mathrm{ps}})^\top\big)\right)^\top$$

$$= \frac{2}{n_{\mathrm{ps}}}(\mathbf{G}\mathbf{Z}_{\mathcal{S}}^{\mathrm{ps}} - \mathbf{X}^{\mathrm{ps}})(\mathbf{Z}_{\mathcal{S}}^{\mathrm{ps}})^\top.$$

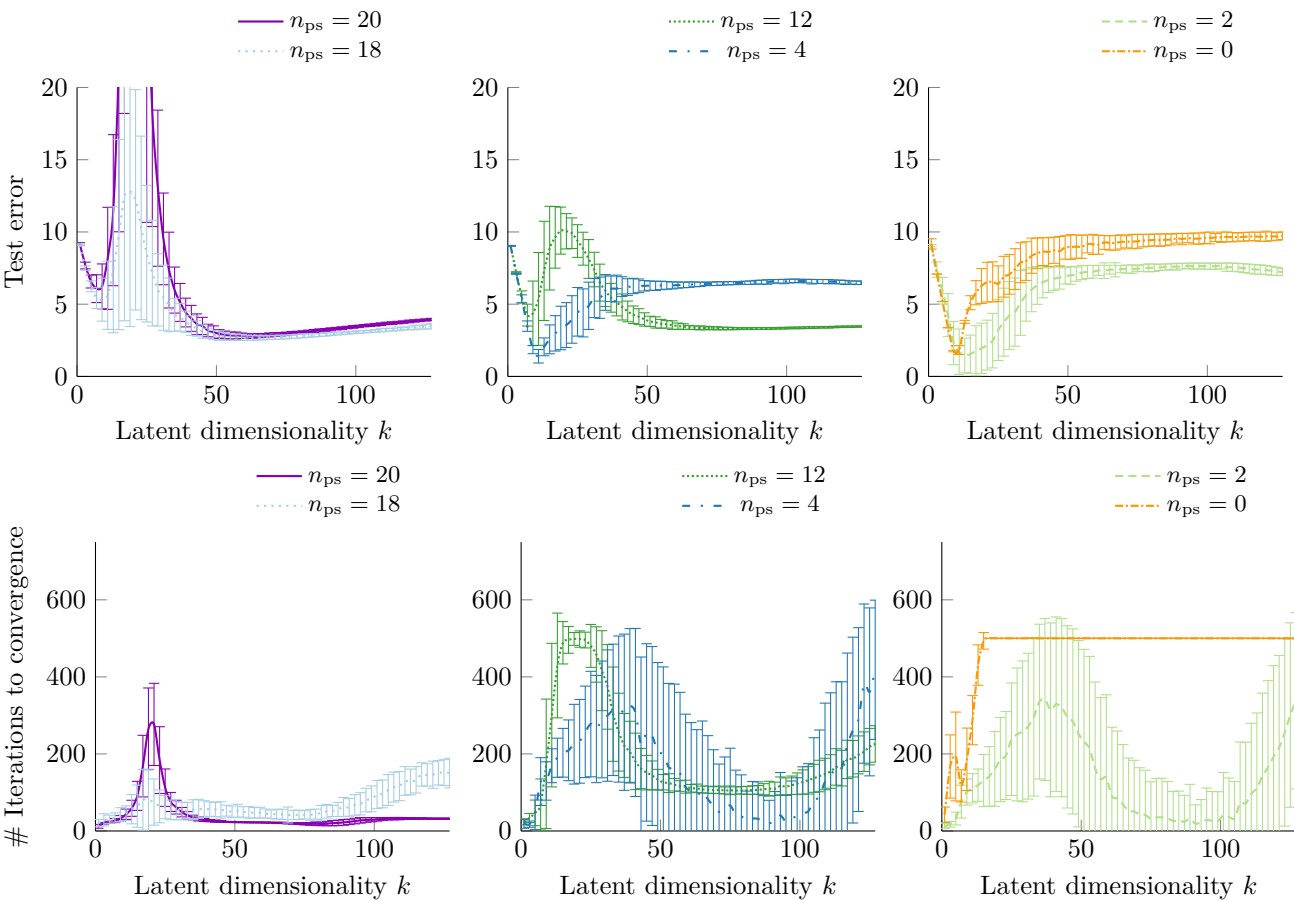

**Figure 9:** In this figure, we minimize the loss in Equation (4). This figure is a more detailed version of the first column of Figure 3. We show results for six different pseudo-supervision levels (i.e., $n_{\mathrm{ps}}$ values). For visual clarity, each subfigure includes results for only two $n_{\mathrm{ps}}$ values. These plots are averaged over 200 experiments, so we plot the standard deviation instead of the standard error (which would be about 14 times smaller).

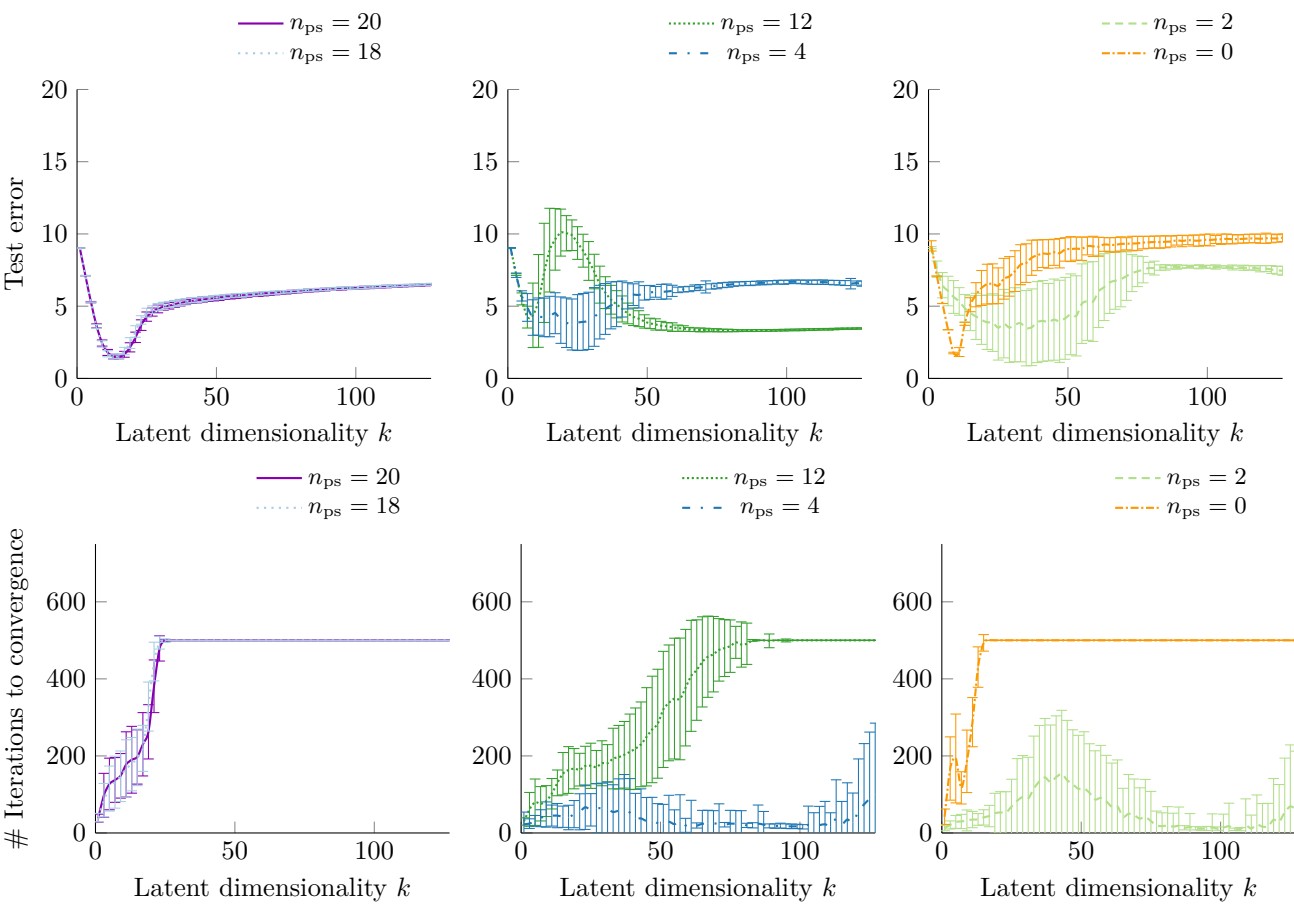

**Figure 10:** In this figure, we minimize the loss in Equation (5). This figure is a more detailed version of the center column of Figure 3. We show results for six different pseudo-supervision levels (i.e., $n_{ps}$ values). For visual clarity, each subfigure includes results for only two $n_{ps}$ values. These plots are averaged over 200 experiments, so we plot the standard deviation instead of the standard error (which would be about 14 times smaller).

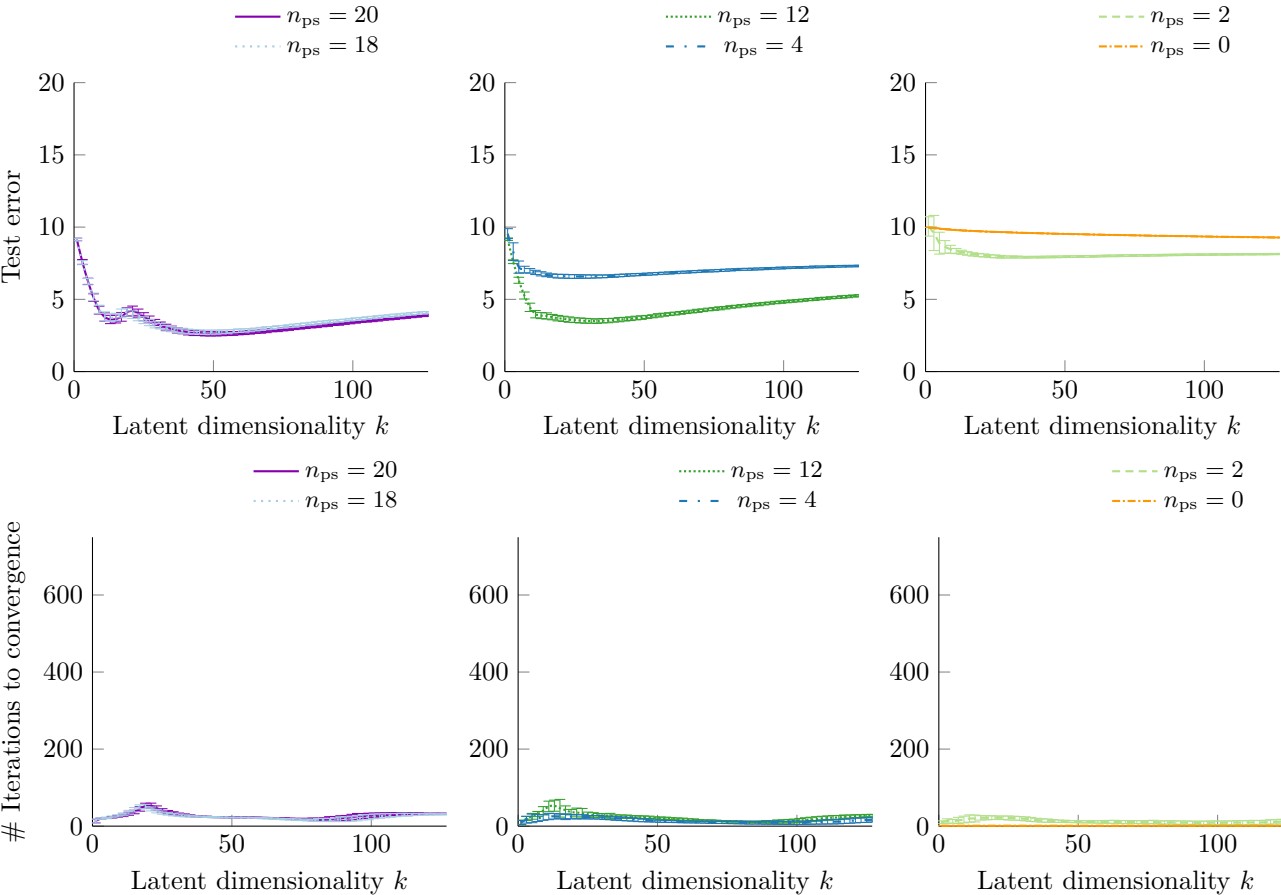

**Figure 11:** In this figure, we minimize the loss in Equation (6). This figure shows that you can achieve good performance and double descent behavior if you weigh the pseudo-supervised and unsupervised terms in the loss disproportionately. Note that in the $n_{\mathrm{ps}} = 0$ case, we are effectively reducing the step size by making $\alpha$ large. In these experiments, we picked $\alpha = 0.98$. We show results for six different pseudo-supervision levels (i.e., $n_{\mathrm{ps}}$ values). For visual clarity, each subfigure includes results for only two $n_{\mathrm{ps}}$ values. These plots are averaged over 200 experiments, so we plot the standard deviation instead of the standard error (which would be about 14 times smaller).

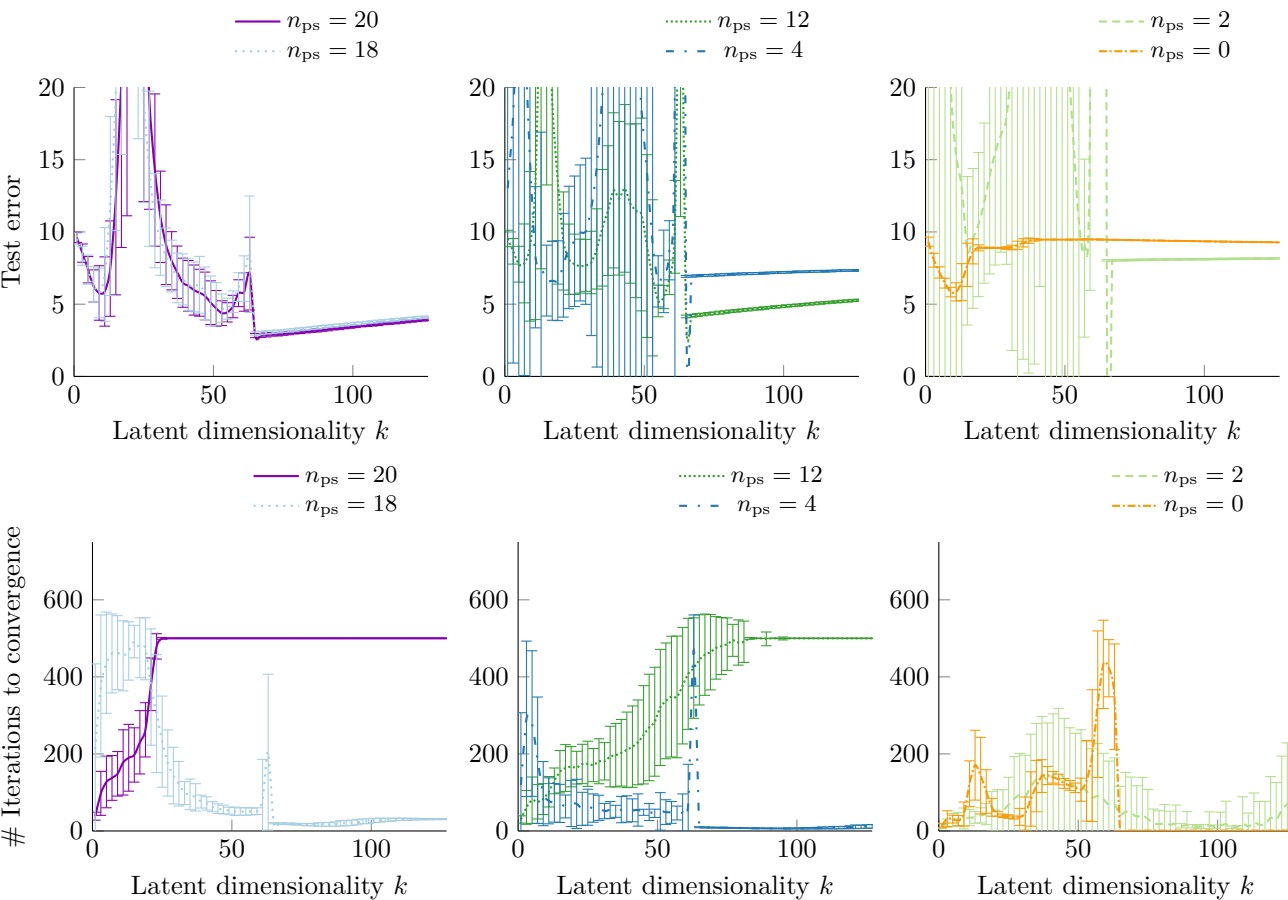

**Figure 12:** In this figure, we minimize the loss in Equation (7). This figure is a more detailed version of the right column of Figure 3. We show results for six different pseudo-supervision levels (i.e., $n_{\text{ps}}$ values). For visual clarity, each subfigure includes results for only two $n_{\text{ps}}$ values. These plots are averaged over 200 experiments, so we plot the standard deviation instead of the standard error (which would be about 14 times smaller).

The gradient of the second term in the considered loss function is a bit more tricky. We simplify it first to get

$$\|(\mathbf{I}_p - \mathbf{G}\mathbf{G}^\top)\mathbf{X}_{\mathcal{S}}^{\mathrm{unsup}}\|_F^2 = \mathrm{Tr}((\mathbf{X}_{\mathcal{S}}^{\mathrm{unsup}})^\top(\mathbf{I}_p - \mathbf{G}\mathbf{G}^\top)(\mathbf{I}_p - \mathbf{G}\mathbf{G}^\top)\mathbf{X}_{\mathcal{S}}^{\mathrm{unsup}})$$
$$= \mathrm{Tr}((\mathbf{X}_{\mathcal{S}}^{\mathrm{unsup}})^\top(\mathbf{I}_p - 2\mathbf{G}\mathbf{G}^\top + \mathbf{G}\mathbf{G}^\top\mathbf{G}\mathbf{G}^\top)\mathbf{X}_{\mathcal{S}}^{\mathrm{unsup}})$$
$$= \|\mathbf{X}_{\mathcal{S}}^{\mathrm{unsup}}\|_F^2 - 2\mathrm{Tr}((\mathbf{X}_{\mathcal{S}}^{\mathrm{unsup}})^\top\mathbf{G}\mathbf{G}^\top\mathbf{X}_{\mathcal{S}}^{\mathrm{unsup}}) + \mathrm{Tr}((\mathbf{X}_{\mathcal{S}}^{\mathrm{unsup}})^\top\mathbf{G}\mathbf{G}^\top\mathbf{G}\mathbf{G}^\top\mathbf{X}_{\mathcal{S}}^{\mathrm{unsup}})$$

which we separate into three terms:

$$f_1(\mathbf{G}) = \|\mathbf{X}_{\mathcal{S}}^{\mathrm{unsup}}\|_F^2$$
$$f_2(\mathbf{G}) = -2\mathrm{Tr}((\mathbf{X}_{\mathcal{S}}^{\mathrm{unsup}})^\top\mathbf{G}\mathbf{G}^\top\mathbf{X}_{\mathcal{S}}^{\mathrm{unsup}})$$
$$f_3(\mathbf{G}) = \mathrm{Tr}((\mathbf{X}_{\mathcal{S}}^{\mathrm{unsup}})^\top\mathbf{G}\mathbf{G}^\top\mathbf{G}\mathbf{G}^\top\mathbf{X}_{\mathcal{S}}^{\mathrm{unsup}}).$$

Clearly, we have that $\nabla_{\mathbf{G}}\|(\mathbf{I}_p - \mathbf{G}\mathbf{G}^\top)\mathbf{X}_{\mathcal{S}}^{\mathrm{unsup}}\|_F^2 = \nabla_{\mathbf{G}}f_1 + \nabla_{\mathbf{G}}f_2 + \nabla_{\mathbf{G}}f_3$ and that $\nabla_{\mathbf{G}}f_1 = 0$. By using some matrix identities, we get that

$$\nabla_{\mathbf{G}}f_2 = -2\nabla_{\mathbf{G}}\mathrm{Tr}((\mathbf{X}_{\mathcal{S}}^{\mathrm{unsup}})^\top\mathbf{G}\mathbf{G}^\top\mathbf{X}_{\mathcal{S}}^{\mathrm{unsup}})$$
$$= -4\mathbf{X}_{\mathcal{S}}^{\mathrm{unsup}}(\mathbf{X}_{\mathcal{S}}^{\mathrm{unsup}})^\top\mathbf{G} \qquad ((119) \text{ from (Petersen \& Pedersen, 2012)})$$

and

$$\nabla_{\mathbf{G}}f_3 = \nabla_{\mathbf{G}}\mathrm{Tr}((\mathbf{X}_{\mathcal{S}}^{\mathrm{unsup}})^\top\mathbf{G}\mathbf{G}^\top\mathbf{G}\mathbf{G}^\top\mathbf{X}_{\mathcal{S}}^{\mathrm{unsup}})$$
$$= \left(\nabla_{\mathbf{G}^\top}\mathrm{Tr}((\mathbf{X}_{\mathcal{S}}^{\mathrm{unsup}})^\top\mathbf{G}\mathbf{G}^\top\mathbf{G}\mathbf{G}^\top\mathbf{X}_{\mathcal{S}}^{\mathrm{unsup}})\right)^\top \qquad (\text{Section 4.2.3 of (Gentle, 2007)})$$
$$= \left(2\mathbf{G}^\top\mathbf{G}\mathbf{G}^\top\mathbf{X}_{\mathcal{S}}^{\mathrm{unsup}}(\mathbf{X}_{\mathcal{S}}^{\mathrm{unsup}})^\top + 2\mathbf{G}^\top\mathbf{X}_{\mathcal{S}}^{\mathrm{unsup}}(\mathbf{X}_{\mathcal{S}}^{\mathrm{unsup}})^\top\mathbf{G}\mathbf{G}^\top\right)^\top \quad ((123) \text{ of (Petersen \& Pedersen, 2012)})$$
$$= 2\mathbf{X}_{\mathcal{S}}^{\mathrm{unsup}}(\mathbf{X}_{\mathcal{S}}^{\mathrm{unsup}})^\top\mathbf{G}\mathbf{G}^\top\mathbf{G} + 2\mathbf{G}\mathbf{G}^\top\mathbf{X}_{\mathcal{S}}^{\mathrm{unsup}}(\mathbf{X}_{\mathcal{S}}^{\mathrm{unsup}})^\top\mathbf{G}$$

Hence, the gradient of the second term in the considered loss function becomes

$$\nabla_{\mathbf{G}}\|(\mathbf{I}_p - \mathbf{G}\mathbf{G}^\top)\mathbf{X}_{\mathcal{S}}^{\mathrm{unsup}}\|_F^2 = \nabla_{\mathbf{G}}f_1 + \nabla_{\mathbf{G}}f_2 + \nabla_{\mathbf{G}}f_3$$
$$= -4\mathbf{X}_{\mathcal{S}}^{\mathrm{unsup}}(\mathbf{X}_{\mathcal{S}}^{\mathrm{unsup}})^\top\mathbf{G} + 2\mathbf{X}_{\mathcal{S}}^{\mathrm{unsup}}(\mathbf{X}_{\mathcal{S}}^{\mathrm{unsup}})^\top\mathbf{G}\mathbf{G}^\top\mathbf{G}$$
$$+ 2\mathbf{G}\mathbf{G}^\top\mathbf{X}_{\mathcal{S}}^{\mathrm{unsup}}(\mathbf{X}_{\mathcal{S}}^{\mathrm{unsup}})^\top\mathbf{G}$$

Thus, the total gradient for Equation (4) becomes

$$\nabla_{\mathbf{G}}\left(\frac{1}{n_{\mathrm{ps}}}\|\mathbf{G}\mathbf{Z}_{\mathcal{S}}^{\mathrm{ps}} - \mathbf{X}^{\mathrm{ps}}\|_F^2 + \frac{1}{n_{\mathrm{unsup}}}\|(\mathbf{I}_d - \mathbf{G}\mathbf{G}^\top)\mathbf{X}^{\mathrm{unsup}}\|_F^2\right) = \frac{2}{n_{\mathrm{ps}}}(\mathbf{G}\mathbf{Z}_{\mathcal{S}}^{\mathrm{sup}} - \mathbf{X}^{\mathrm{sup}})(\mathbf{Z}_{\mathcal{S}}^{\mathrm{sup}})^\top$$
$$- \frac{4}{n_{\mathrm{unsup}}}\mathbf{X}^{\mathrm{unsup}}(\mathbf{X}^{\mathrm{unsup}})^\top\mathbf{G}$$
$$+ \frac{2}{n_{\mathrm{unsup}}}\mathbf{X}^{\mathrm{unsup}}(\mathbf{X}^{\mathrm{unsup}})^\top\mathbf{G}\mathbf{G}^\top\mathbf{G}$$
$$+ \frac{2}{n_{\mathrm{unsup}}}\mathbf{G}\mathbf{G}^\top\mathbf{X}^{\mathrm{unsup}}(\mathbf{X}^{\mathrm{unsup}})^\top\mathbf{G}.$$

The gradient for the loss in Equation (7) is similar. Again, we restate the loss for completeness:

$$\mathcal{L}^{\mathrm{train}}(\mathbf{G}, \mathcal{D}) = \frac{1}{n_{\mathrm{ps}}}\|\mathbf{G}\mathbf{Z}_{\mathcal{S}}^{\mathrm{ps}} - \mathbf{X}^{\mathrm{ps}}\|_F^2 + \frac{1}{n}\|(\mathbf{I}_d - \mathbf{G}\mathbf{G}^\dagger)\mathbf{X}\|_F^2$$

The gradient of the first term of Equation (7) is the same as in the above result for the loss in Equation (4). The gradient of the second term of Equation (7) requires more work. With $\mathbf{B} = \mathbf{X}\mathbf{X}^\top$ for shorthand and

assuming that $\mathbf{G}$ has full column rank, we see that

$$
\begin{aligned}
\nabla_{\mathbf{G}}\|(\mathbf{I}_d - \mathbf{G}\mathbf{G}^\dagger)\mathbf{X}\|_F^2 &= \nabla_{\mathbf{G}}\mathrm{Tr}((\mathbf{I}_d - \mathbf{G}\mathbf{G}^\dagger)(\mathbf{I}_d - \mathbf{G}\mathbf{G}^\dagger)\mathbf{B}) \\
&= \nabla_{\mathbf{G}}\mathrm{Tr}((\mathbf{I}_d - 2\mathbf{G}\mathbf{G}^\dagger + \mathbf{G}\mathbf{G}^\dagger\mathbf{G}\mathbf{G}^\dagger)\mathbf{B}) \\
&= \nabla_{\mathbf{G}}\mathrm{Tr}((\mathbf{I}_d - 2\mathbf{G}\mathbf{G}^\dagger + \mathbf{G}\mathbf{G}^\dagger)\mathbf{B}) \\
&= \nabla_{\mathbf{G}}\mathrm{Tr}((\mathbf{I}_d - \mathbf{G}\mathbf{G}^\dagger)\mathbf{B}) \\
&= \nabla_{\mathbf{G}}\mathrm{Tr}(\mathbf{B}) - \nabla_{\mathbf{G}}\mathrm{Tr}(\mathbf{G}\mathbf{G}^\dagger\mathbf{B}) \\
&= -\nabla_{\mathbf{G}}\mathrm{Tr}(\mathbf{G}(\mathbf{G}^\top\mathbf{G})^{-1}\mathbf{G}^\top\mathbf{B}) \\
&= -\nabla_{\mathbf{G}}\mathrm{Tr}((\mathbf{G}^\top\mathbf{G})^{-1}\mathbf{G}^\top\mathbf{B}\mathbf{G}) \\
&= -\nabla_{\mathbf{G}}\mathrm{Tr}((\mathbf{G}^\top\mathbf{G})^{-1}\mathbf{G}^\top\mathbf{B}\mathbf{G}) \\
&= 2\mathbf{G}(\mathbf{G}^\top\mathbf{G})^{-1}\mathbf{G}^\top\mathbf{B}\mathbf{G}(\mathbf{G}^\top\mathbf{G})^{-1} - 2\mathbf{B}\mathbf{G}(\mathbf{G}^\top\mathbf{G})^{-1}. \quad \text{((126) in (Petersen \& Pedersen, 2012))}
\end{aligned}
$$

Thus, the total gradient for Equation (7) becomes

$$
\begin{aligned}
\nabla_{\mathbf{G}}\left(\frac{1}{n_{\mathrm{ps}}}\|\mathbf{G}\mathbf{Z}_{\mathcal{S}}^{\mathrm{ps}} - \mathbf{X}^{\mathrm{ps}}\|_F^2 + \frac{1}{n}\|(\mathbf{I}_d - \mathbf{G}\mathbf{G}^\dagger)\mathbf{X}\|_F^2\right) &= \frac{2}{n_{\mathrm{ps}}}(\mathbf{G}\mathbf{Z}_{\mathcal{S}}^{\mathrm{ps}} - \mathbf{X}^{\mathrm{ps}})(\mathbf{Z}_{\mathcal{S}}^{\mathrm{ps}})^\top \\
&\quad + \frac{2}{n}\mathbf{G}(\mathbf{G}^\top\mathbf{G})^{-1}\mathbf{G}^\top\mathbf{X}\mathbf{X}^\top\mathbf{G}(\mathbf{G}^\top\mathbf{G})^{-1} \\
&\quad - \frac{2}{n}\mathbf{X}\mathbf{X}^\top\mathbf{G}(\mathbf{G}^\top\mathbf{G})^{-1}.
\end{aligned}
$$

If $\mathbf{G}$ has full row rank instead, one gets a similar gradient expression. During the minimization of the loss in Equation (7), the matrix $\mathbf{G}$ may become close to low rank and make the gradient calculation unstable. For numerical stability of the gradient, we calculate $(\mathbf{G}^\dagger)^\top$ instead of $\mathbf{G}(\mathbf{G}^\top\mathbf{G})^\dagger$.

## E  Experiments on nonlinear, multilayer GANs on MNIST

In this section we provide details for the experiments on nonlinear, multilayer GANs. One of these experiments is discussed in Section 5 and the other is an additional experiment which is not in the main paper. The details here are relevant to both experiments.

We train a gradient penalized Wasserstein GAN (WGAN-GP) (Gulrajani et al., 2017) on MNIST (LeCun et al., 1998). The architecture output directly from PyTorch is shown below with the latent dimensionality changed to $k$, as it varies in our experiments:

```
Generator(
  (model): Sequential(
    (0): Linear(in_features=k, out_features=128, bias=True)
    (1): LeakyReLU(negative_slope=0.2, inplace)
    (2): Linear(in_features=128, out_features=256, bias=True)
    (3): BatchNorm1d(256, eps=0.8, momentum=0.1, affine=True, track_running_stats=True)
    (4): LeakyReLU(negative_slope=0.2, inplace)
    (5): Linear(in_features=256, out_features=512, bias=True)
    (6): BatchNorm1d(512, eps=0.8, momentum=0.1, affine=True, track_running_stats=True)
    (7): LeakyReLU(negative_slope=0.2, inplace)
    (8): Linear(in_features=512, out_features=1024, bias=True)
    (9): BatchNorm1d(1024, eps=0.8, momentum=0.1, affine=True, track_running_stats=True)
    (10): LeakyReLU(negative_slope=0.2, inplace)
    (11): Linear(in_features=1024, out_features=784, bias=True)
    (12): Tanh()
  )
)
```

```
Discriminator(
  (model): Sequential(
    (0): Linear(in_features=784, out_features=512, bias=True)
    (1): LeakyReLU(negative_slope=0.2, inplace)
    (2): Linear(in_features=512, out_features=256, bias=True)
    (3): LeakyReLU(negative_slope=0.2, inplace)
    (4): Linear(in_features=256, out_features=1, bias=True)
  )
)
```

Note that we control the parameterization of our GAN by modifying just the latent dimension. However, since these networks are multi-layer, one could investigate the behavior of a GAN by varying the widths of all the layers at once or even one layer at a time. This is an interesting topic which merits further exploration in future work.

The networks are trained with a gradient penalty weight of $\lambda_{\mathrm{GP}} = 10$. The pseudo-supervised sample pairs were fixed as we varied $k$ so that the plots were comparable. However, we ran both of these experiments over 10 trials, with 10 sets of pseudo-supervised samples corresponding to 10 subsets of the training data. Let us denote $\mathcal{L}_{\mathrm{GP}}$ as the WGAN-GP objective function. We trained the discriminator as usual, and trained the generator with the following modified objective function:

$$\mathcal{L}_G(\mathbf{X}^{\mathrm{batch}}, \mathbf{X}^{\mathrm{ps}}, \mathbf{Z}^{\mathrm{ps}}) = \mathcal{L}_{\mathrm{GP}}(\mathbf{X}^{\mathrm{batch}}) + \|G(\mathbf{Z}^{\mathrm{ps}}) - \mathbf{X}^{\mathrm{ps}}\|_F^2$$

with $\mathbf{X}^{\mathrm{batch}} \in \mathbb{R}^{d \times n_{\mathrm{batch\ size}}}, \mathbf{X} \in \mathbb{R}^{d \times n_{\mathrm{ps}}}$, and $\mathbf{Z}^{\mathrm{ps}} \in \mathbb{R}^{k \times n_{\mathrm{ps}}}$ for generator $G$. Additionally, one could weigh this pseudo-supervised term more or less, however we found that a weight of 1 was adequate to get our results.

For all of our experiments with nonlinear, multilayer GANs, we have a batch size of 4096, an ADAM learning rate of 0.0002, ADAM hyperparameters $\beta = (0.5, 0.999)$, and clip value of 0.01. We train the discriminator 5 times per iteration. The optimizer values are used for both generator and discriminator. In the main paper, we train for 3000 iterations and use 4096 total samples from the training data so that we are performing gradient descent instead of stochastic gradient descent (SGD). We also do experiments for SGD in Appendix E.1.

We measure test error with geometry score (Khrulkov & Oseledets, 2018) as it is better suited for MNIST than other performance measures, such as Fréchet Inception Distance (Heusel et al., 2017) and Inception Score (Salimans et al., 2016) which are better suited for natural images. For our calculation of the geometry score, we pick $L_0 = 32, \gamma = \frac{1}{1000}, i_{\max} = 100$, and $n = 100$ as done in the original paper when computing scores for the MNIST dataset. Moreover, we generate 10,000 images and compare these generated images to the MNIST test set, which also contains 10,000 images.

In Figure 13 we provide errorbars for Figure 4 to show that pseudo-supervision lowers variance.

## E.1   Pseudo-supervision with stochastic gradient descent

In this section, we train a WGAN-GP just as above except for two changes: we train using all the training data (60000 samples) for 200 iterations using SGD. We only train for 200 iterations because now each epoch has about 15 batches instead of the single batch in the previous section.

Our results are shown in Figure 14 and Figure 15. We see that with SGD, we lose the double descent but consistently beat the baseline. We also converge faster than the baseline, but not as fast as the pure gradient descent setting. Moreover, we reduce the variance in the test error across experiments drastically compared to the baseline.

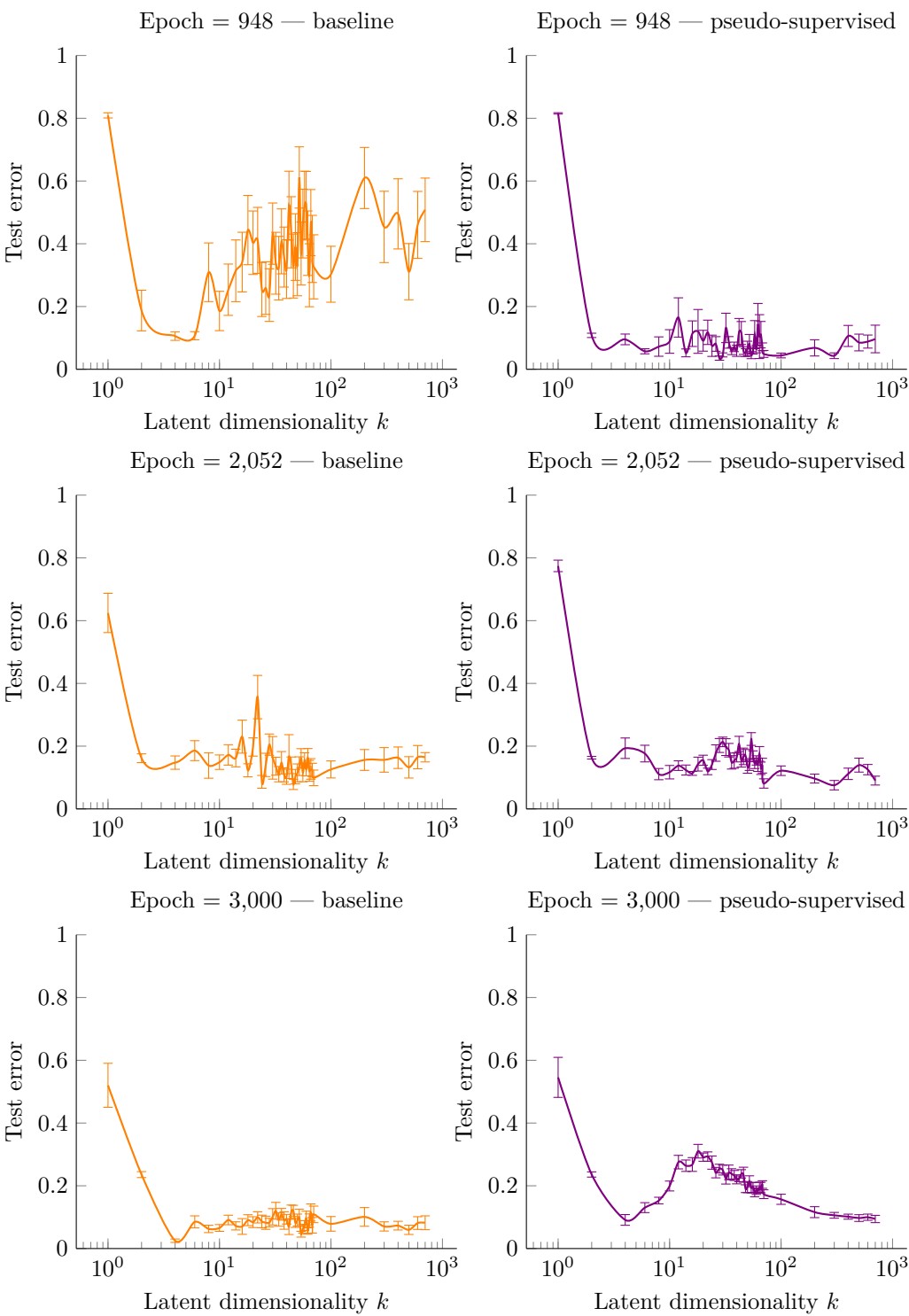

**Figure 13:** In this figure, we train a WGAN-GP on MNIST using gradient descent on a subset of 4096 training images. This figure is a more detailed version of Figure 4. . We plot the standard errors as errorbars.

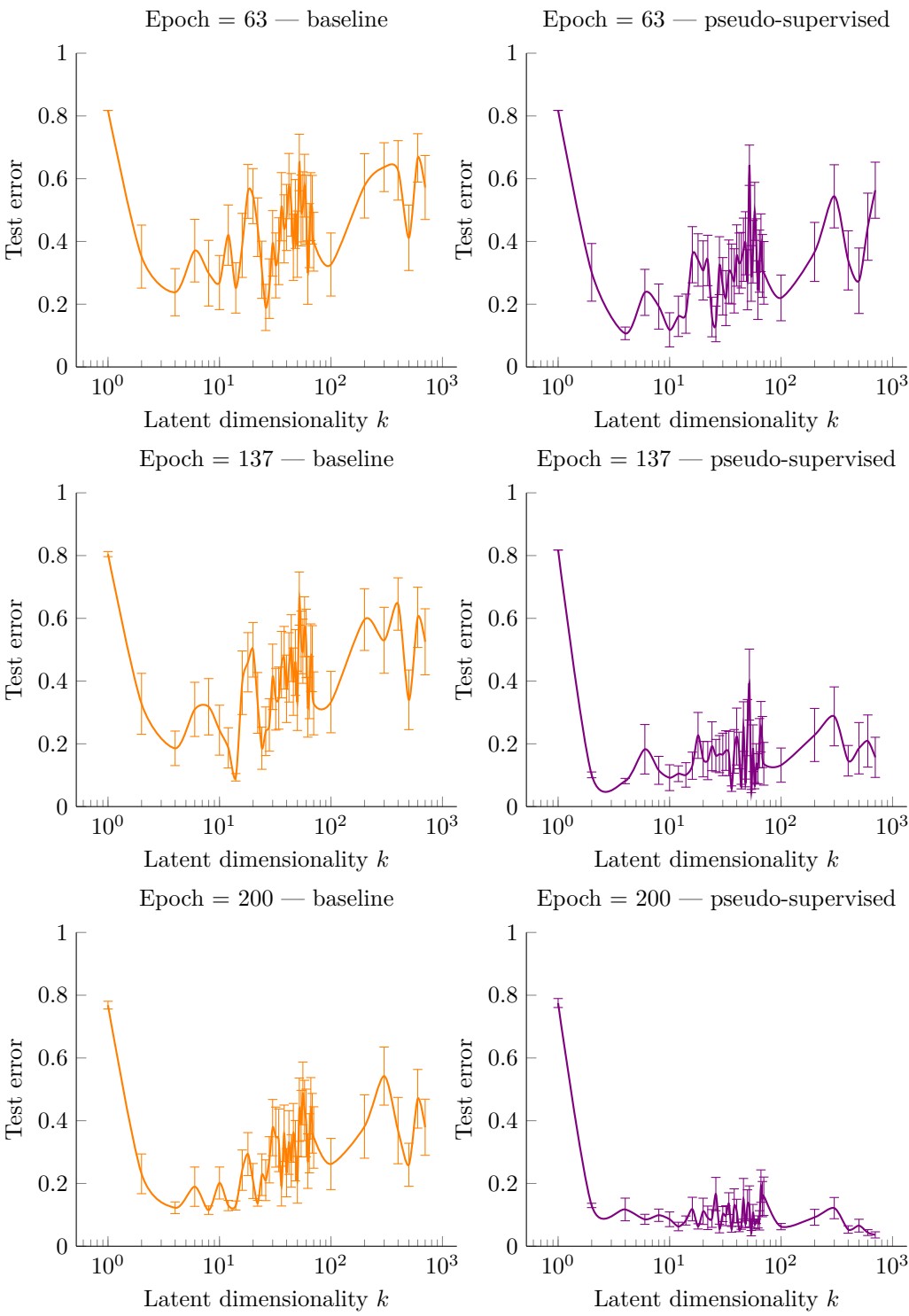

**Figure 14:** In this figure, we train a WGAN-GP on the full MNIST dataset using SGD. The pseudo-supervised GAN has much lower variance and outperforms the baseline later in training. Convergence speed is also faster for the pseudo-supervised model. We plot the standard errors as errorbars.

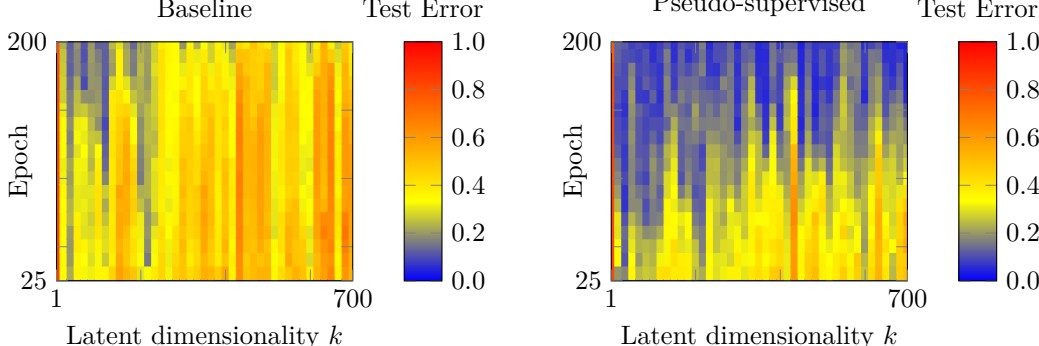

**Figure 15:** In this figure, we train a WGAN-GP on the full MNIST dataset using SGD. The pseudo-supervised GAN converges to a low error much faster than the baseline. Just as in Figure 5, each point in the heatmap is an average test error over 10 networks The test error is measured by geometry score here. The $k$-axis is plotted so that each column corresponds to the next entry for better visualization, even though the spacing is $k \in \{1, 2, 4, 6, \ldots, 70, 100, 200, 300, \ldots, 700\}$.

