# OpenReview forum: "Double Descent and Other Interpolation Phenomena in GANs"
_TMLR — Rejected by TMLR_

### Review · Reviewer_HckF · 2024-05-03

**Summary Of Contributions:**

This paper studies the generalization of GAN-based algorithms and proposes a pseudo-supervised learning approach to improve the generalization in the over-parametrized regime.

**Audience:**

Yes

**Claims And Evidence:**

No

**Requested Changes:**

Please check the weaknesses.

**Strengths And Weaknesses:**

[1] Proof of Observation 1 needs more explanation: the last sentence "By letting..." assigns a value to L_{interpolate}^{test} instead of proving that L_{interpolate}^{test}>0. Also, even if L_{interpolate}^{test}>0, it is possible that L_{interpolate}^{test}->0 in some regimes. The current proof does not screen out this case.

[2] There is no proof for Corollary 1.

[3] While the formulas in this paper are mainly for linear models, there is no formal theoretical results describing any exact observation in the testing loss in such a scenario. Although Theorem 1 and 2 provides some theotical results, they are not sufficient. Please follow double-descent literature, e.g.,

Belkin, Mikhail, Daniel Hsu, and Ji Xu. "Two models of double descent for weak features." SIAM Journal on Mathematics of Data Science 2.4 (2020): 1167-1180.

Hastie, Trevor, et al. "Surprises in high-dimensional ridgeless least squares interpolation." Annals of statistics 50.2 (2022): 949.

Kausik, Chinmaya, Kashvi Srivastava, and Rishi Sonthalia. "Double Descent and Overfitting under Noisy Inputs and Distribution Shift for Linear Denoisers." Transactions on Machine Learning Research.

to formally prove the claims in linear models.

[4] A confusion on double descent. It is not clear to me why the authors highlight double descent phenomenon and try to craft a double-descent result for their proposed method. Based on my understanding, double-descent phenomenon is an outcome of some ML methods given certain data, and it is not a purpose of a ML model. In this paper, my understanding is that the authors observe a poor generalization for GAN-based models, and propose a method to improve generalization in the over-parameterized regime. There is no need to mention "double descent".

[5] The curves in the experiments fluctuate a lot. Please consider repeating the experiments for more times and obtain an average (i.e., a smoother curve).

---

> ### Author Response · Authors · 2024-06-23
> **Response to reviewer**
>
> Thank you for your feedback and comments on our work. Below we respond to your concerns, let us know if you have any questions regarding our answers.
> 1. As you said, the last sentence does assign a value to $L_{interpolate}^{test}$. We do not claim that $L_{interpolate}^{test} > 0$ but rather that $L_{interpolate}^{test} \geq 0$, which is a direct consequence of $q$ being an $f$-divergence or metric. The properties of metrics can be found, for example, in [1]. Additionally, you asked if $L_{interpolate}^{test}$ can approach $0$ in some circumstances; yes, it can. It can also be $0$ directly if, for example, $p_t = p_f$, which is implied with the $\geq$.
> 2. You asked about a proof for Corollary 1. The sentence following the corollary explains the reasoning of why the corollary is true. “There is no double descent behavior because the test error is constant in the overparameterized regime of interpolating solutions.” We chose to call this result a corollary becauase, typically, corollaries do not require proofs (as opposed to theorems, lemmas, and propositions) as they are immediate consequences of the preceding content.
> 3. You mention that we do not have enough theoretical results in our work and point to Theorem 1 and 2. We would like to point out that we also have theorems 3 and 4 later in the paper. Additionally, you correctly state that we do not describe the exact behavior of the test error as those other works do. We agree that the test error formulation would be a great future direction. However, the acceptance criteria document (https://jmlr.org/tmlr/acceptance-criteria.html) states that the most important factor in determining paper acceptance is if we support our claims with sufficient evidence. We do not need to provide exact formulations for the test error because we do not make any claims about knowing the exact formulation of the test error.
> 4. We mention double descent for several reasons; one of them is that we observe double descent in several settings in which double descent has not been previously observed.
> 5. Thank you for pointing out that our curves fluctuate a little too much. It turns out that we were plotting the standard deviation instead of the standard error by mistake and we have since corrected our mistake.
>
> We hope that you find our explanations here adequate. Let us know if there is anything else that we can do to improve our paper.
>
> [1] Rudin, Walter. Principles of mathematical analysis. Vol. 3. New York: McGraw-hill, 1964.

---

### Review · Reviewer_QjSy · 2024-06-03

**Summary Of Contributions:**

This paper studies the double descent phenomenon when training a linear generative model theoretically and vanilla GAN empirically, which suggests that pseudo-supervision can be used to achieve beneficial double descent phenomena in unsupervised models.

**Audience:**

Yes

**Broader Impact Concerns:**

There are not any concerns on the ethical implications of the work.

**Claims And Evidence:**

Yes

**Requested Changes:**

There are some changes I suggest
- Present more derivations to show the connection between linear GAN and PCA as in Eq. (2).
- Give more details regarding how to compute the training error and testing error in Figures 1,2, and 3.

**Strengths And Weaknesses:**

## Strengths
- Some theories are developed to characterize the zero-loss solutions for the relevant objective functions in unsupervised and pseudo-supervision of linear generative models.
- The experiments in Section 5 are intensive because they are conducted with 430 GAN models for various $k$.

## Weaknesses
- Some parts in the paper are not very clear.
  (1) The connection between linear GAN and PCA shown in Eq. (2). The authors should strengthen this part with more derivations and explanation. It seems that it is using the maximum likelihood for training linear GAN, hence leading to the formulation of PCA?
  (2) It is not very clear about the training error and testing error in Figures 1,2, and 3 although the paper has briefly mentioned 2-Wasserstein distance. However, it is unclear of two distributions under calculation (e.g., the generative Gaussian distribution and what distribution).
- The theory part revolves a linear generative model with Gaussian distributions. It is hard to say that it is generative adversarial network because there is not any adversarial mechanism in training. Therefore, it is not indeed a linear GAN.

---

> ### Author Response · Authors · 2024-06-23
> **Response to reviewer**
>
> Thank you for your feedback and comments on our work. Below we respond to your concerns, let us know if you have any questions about our answers.
> 1. The connection between linear GANs and PCA was shown in [1] and we will explain a bit here. They studied GANs with linear generators, quadratic discriminators, and Gaussian data (this has been named the LQG setting). This result is not our work so that is why we don’t spend too much time on it. We do state the following in the background, hopefully it is clear what we mean: ``One result in the LQG setting [1] is that the principal component analysis (PCA) solution is an optimal solution for the generator in the minmax optimization.”
> 2. Your concern about how the train and test errors are calculated is a very good point. We added the following to the end of the figure captions: ``The train errors (left subfigure) and test errors (right subfigure) are calculated with the $2$-Wasserstein metric; the training error is between the training data and the PCA estimate and the test error is between the true distribution and the PCA estimate.’’
>
> We hope that this addresses your concerns. Let us know if there is anything else that we can do.
>
> [1] Feizi, Soheil, et al. "Understanding GANs in the LQG setting: Formulation, generalization and stability." IEEE Journal on Selected Areas in Information Theory 1.1 (2020): 304-311.

---

### Review · Reviewer_8uKJ · 2024-06-09

**Summary Of Contributions:**

This work studies overparameterization in generative adversarial networks (GANs) that can interpolate the training data. The main contributions are:
* Analysis of generalizations of linear GANs at different variations of the latent space dimensions.
* They observe that a training process minimizing a distribution metric in GANs have same performance of all the overparameterized solutions.
* To improve the generalization performance in overparameterized setting a pseudo-supervised training approach is taken.  This encourages double-descent in linear GANs. The results are also further extended to non-linear GANs for the binary MNIST dataset.

**Audience:**

Yes

**Broader Impact Concerns:**

None.

**Claims And Evidence:**

Yes

**Requested Changes:**

* The paper should better motivate their choice of evaluation of datasets. The paper should also consider including experiments with the full MNIST or CelebA datasets.

* Pseudo-supervision label selection process should be described in more detail.

* The relationship between over-parameterization in terms of latent dimensions versus model parameters should be explained in more detail.

**Strengths And Weaknesses:**

Strengths:
* The work is clearly written and shows the benefits of double descent phenomenon for unsupervised GANs.

* The proposed pseudo-supervision is shown to accelerate training and lower the generalization error.

* This is one of the first works to explore the effect of over-parameterization in unsupervised models like GANs.

Weakness:
* The results are demonstrated only on the binary MNIST dataset. Given that it is a 784 dimensional, in practice it is easy to achieve convergence in MNIST. These results do not necessarily translate to other datasets. Considering 128 datapoints for CelebA is not realistic.

* The proposed method requires the pseudo-supervision labels to be fabricated. What would be the effect of scaling the method to datasets with millions or billions of data points, as it is the case with modern datasets.

* The paper considers over-parameterization only in terms of latent dimensions. The effect of model parameters is not considered.

* The experiments are performed only with Wasserstein GANs. Other formulations such as with JS divergence or MMD are not considered.

---

> ### Author Response · Authors · 2024-06-23
> **Response to reviewer**
>
> Thank you for your feedback and comments on our work. Below we respond to your concerns, let us know if you have any questions about our answers.
> 1. We have considered using the whole dataset but we run into issues doing this. We have added the following explanation to our paper in Section 5:
> ``In addition to the reasons stated above, for the MNIST and CelebA experiments, we use subsets of data because if we use the whole dataset we run out of orthogonal pseudo-supervised vectors. This is a limitation of our work when it comes to smaller networks which have small latent spaces compared to the number of data points.’’
> 2. We define in our paper (at the end of Section 4.1) how the pseudo-supervised vectors are drawn. We added this sentence to clarify:
> ``Therefore, the pseudo-supervised vectors are independent and identically distributed samples from an isotropic Gaussian distribution.’’
> 3. Thank you for recommending that we use the total model parameters to determine complexity rather than just the latent dimension. This is an interesting direction for future work and we added this to Section 5 of the paper: ``Note that we control the parameterization of our GAN by modifying just the latent dimension. However, since these networks are multi-layer, one could investigate the behavior of a GAN by varying the widths of all the layers at once or even one layer at a time. This is an interesting topic which merits further exploration in future work.''
>
> We hope that this addresses your concerns. Let us know if there is anything else that we can do.

---

### Decision · Action_Editor_CtHs · 2024-08-02

**Recommendation:** Reject

**Comment:**

The paper studies overparameterization in generative adversarial networks (GANs). The authors demonstrate that overparameterization can enhance generalization performance and speed up training. They examine generalization errors relative to latent space dimensions and make two claims:

1. Overparameterized GANs learning distributions by minimizing a metric or divergence do not show double descent in generalization errors.
2. A novel pseudo-supervised learning approach exhibits double descent in generalization errors. This approach, when combined with overparameterization, accelerates training and matches or surpasses generalization performance without pseudo-supervision.

The reviewers highlighted strengths like the clear writing and innovative pseudo-supervision approach, but also pointed out weaknesses such as limited dataset applicability and scalability issues. Despite some empirical and theoretical contributions, the paper received mixed reviews, with concerns about clarity, insufficient empirical evidence, and shallow theoretical proofs. The decision of rejection is mainly based on the insufficient empirical evidence, done on only two sub-sampled benchmark dataset and one specific type of GAN. While the authors are appreciated for running over 400 GAN models in the experiments, claim 2 is at the moment only true for the sub-sampled data when "enough orthogonal pseudo-supervised vectors can be found" (as the authors' rebuttal states), and the general version of the claim is not sufficiently supported.

**Audience:**

Yes.

**Claims And Evidence:**

No. Experiments are insufficient to support the claim (see comments).